ecology/genetics

Arabian oryx, *Oryx leucoryx*, Restriction site-associated DNA sequencing (RAD-sequencing), Double digest restriction-site associated DNA (ddRADseq), single nucleotide polymorphisms, population genetics, genotyping

**Author for correspondence:**
Jaime Gongora
e-mail: jaime.gongora@sydney.edu.au

# Rescued back from extinction in the wild: past, present and future of the genetics of the Arabian oryx in Oman

Qais Al Rawahi[1,2,3], Jose Luis Mijangos[1,5], Mehar S. Khatkar[1], Mohammed A. Al Abri[6], Mansoor H. AlJahdhami[2], Jennifer Kaden[4], Helen Senn[4], Katherine Brittain[1] and Jaime Gongora[1]

[1]Sydney School of Veterinary Science, Faculty of Science, The University of Sydney, Sydney, NSW 2006, Australia
[2]Office for Conservation of the Environment, Diwan of Royal Court, PO Box 246, P.C. 100, Muscat, Oman
[3]College of Applied Sciences, A'Sharqiyah University, PO Box 42, Postal Code 400, Ibra, Sultanate of Oman
[4]RZSSWildGenes Laboratory, Royal Zoological Society of Scotland, Edinburgh EH12 6TS, UK
[5]Centre for Conservation Ecology and Genomics, Institute for Applied Ecology, University of Canberra, Canberra, ACT, 2617, Australia
[6]Department of Animal and Veterinary Sciences, College of Agricultural and Marine Sciences, Sultan Qaboos University, Muscat, Oman

MAAA, 0000-0003-1708-3845; HS, 0000-0002-3711-8753; KB, 0000-0003-0000-6499

The Arabian oryx was the first species to be rescued from extinction in the wild by the concerted efforts of captive programmes in zoos and private collections around the world. Reintroduction efforts have used two main sources: the 'World Herd', established at the Phoenix Zoo, and private collections in Saudi Arabia. The breeding programme at the Al-Wusta Wildlife Reserve (WWR) in Oman has played a central role in the rescue of the oryx. Individuals from the 'World Herd' and the United Arab Emirates have been the main source for the WWR programme. However, no breeding strategies accounting for genetic diversity have been implemented. To address this, we investigated the diversity of the WWR population and historical samples using mitochondrial DNA (mtDNA) and single nucleotide polymorphisms (SNPs). We found individuals at WWR contain 58% of the total mtDNA diversity observed globally. Inference of ancestry and spatial patterns of SNP variation shows the presence of three ancestral sources and three different groups of individuals. Similar levels of diversity and low inbreeding were observed between groups. We identified

individuals and groups that could most effectively contribute to maximizing genetic diversity. Our results will be valuable to guide breeding and reintroduction programmes at WWR.

# 1. Introduction

The Arabian oryx (*Oryx leucoryx*) is currently described as 'vulnerable' by the International Union for Conservation of Nature (IUCN) Red List of Threatened Species [1]. It is an ungulate [2] that became extinct in the wild in 1972 [3]. However, conservation efforts rescued the Arabian oryx from a complete extinction through the establishment of successful breeding and reintroduction programmes in the Arabian Peninsula [4]. Conservation efforts in other parts of the world included the establishment of the 'World Herd' in the Phoenix Zoo (USA) using the last remaining wild individuals which were captured during 'Operation Oryx' in 1962 and those donated by Arabian rulers [5] consisting of nine individuals from Oman, Kingdom of Saudi Arabia (KSA) and Yemen. A subsequent breeding group was established in Los Angeles Zoo (USA) using a pair of individuals from Riyadh Zoo (KSA). Through the concerted breeding efforts carried out by various USA institutions, the 'World Herd' increased to 105 individuals, and by 1978, some of these individuals were translocated to the Middle East and Europe. The current number of the Arabian oryx is estimated to be approximately 1100 in the wild and approximately 6000–7000 in captivity, which includes zoos, reserves and private collections [1]. The history of the management of the Arabian oryx is a complex one involving a series of reintroductions, supplemental releases and transfers of individuals between the different captive populations in the Arabian Peninsula. In some cases, management details were not recorded (or published), and captive breeding was not implemented using individual genetic management through minimizing population mean kinship.

Previous genetic studies have found relatively low levels of population differentiation and substantial genetic mixing between the major Arabian oryx groups that have been used in the different conservation programmes across the world [6–10]. More recent microsatellite studies of current captive breeding programmes in Jordan, the United Arab Emirates (UAE) and Qatar found relatively low levels of genetic diversity (e.g. observed heterozygosity ranging from 0.332 to 0.535), some differentiation between herds or groups and a high level of inbreeding and relatedness (e.g. inbreeding coefficient (FIS) = 0.073–0.431; electronic supplementary material, table S1; [6,7,10–13]). By contrast, Arif *et al.* [14] found high levels of genetic diversity in a Saudi Arabian captive group. The low levels of genetic diversity reported in these studies might be the result of a period of isolation over multiple generations between the main founder groups. In addition to the research completed using microsatellites, 12 mitochondrial DNA control region (mtDNA CR) haplotypes were identified in the Arabian oryx populations (electronic supplementary material, tables S1 and S2; [11,12,15]). The above genetic studies have provided important information to develop basic herd-management recommendations; however, they do not have the resolution required to inform more intensive individual-based management or individual breeding strategies [7].

A reintroduction and captive breeding programme for the Arabian oryx was established at the Al-Wusta Wildlife Reserve (WWR) in Jalooni in Oman in the 1980s using individuals from the 'World Herd' (figure 1). Eventually, through successful management, the Jalooni population increased to about 400 individuals in the wild by 1996 [16]. However, due to excessive and persistent poaching, the wild population soon dropped sharply to 108 males and 28 females by 1998 [16]. Action was quickly taken to rescue the remaining wild individuals at Jalooni by bringing them back into captivity and establishing a new captive breeding programme with 100 founder individuals (referred in this paper as the WWR-Oman group or herd). To increase size and diversity of the population in the WWR programme, 100 additional individuals from the UAE, which had a mixed ancestry from different sources, including the 'World Herd' and private collections in UAE [18], were translocated into the WWR in 2009 (referred in this paper as the WWR-UAE group or herd). This has resulted in a population of 650 individuals at the WWR to date. The breeding programme at WWR has focused on increasing population size rather than increasing genetic diversity; therefore, individuals from the two source populations (WWR-Oman and WWR-UAE) were allowed to mate randomly. Offspring of these two source populations is referred in this paper as the WWR-Mix group or herd. Animals at WWR were divided randomly irrespective of their source population and/or parental relationships and have been kept in three enclosures (ranging from 0.27 to 0.74 km$^2$); however, individuals are continuously rotated at random between the three enclosures. Since the WWR programme did not implement a specific genetic management strategy, studbook information and pedigree records are not available. The lack of this information poses various challenges for the management of this population, especially because one of its major sources, the

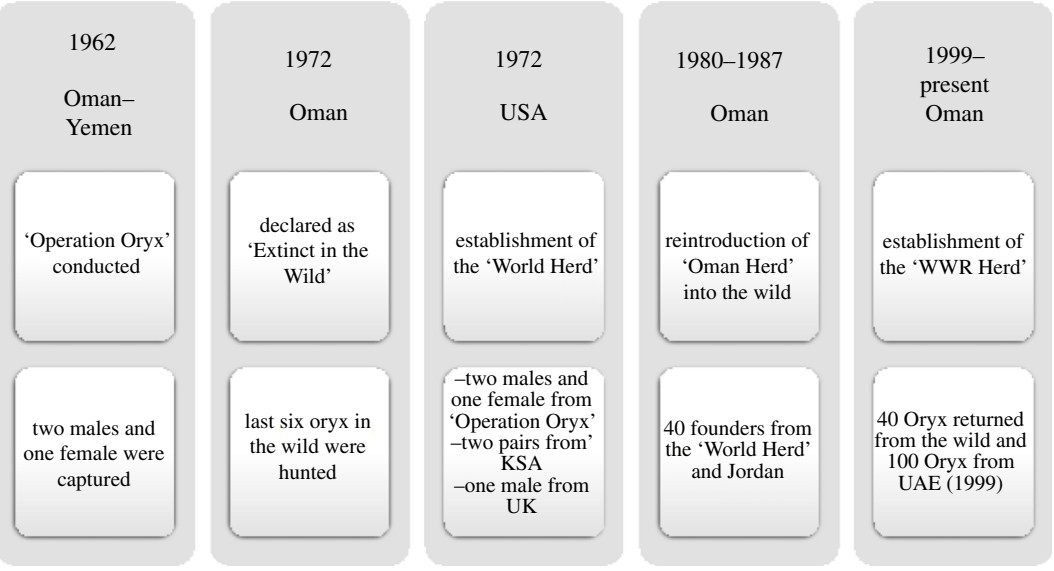

**Figure 1.** Timeline of the Arabian oryx reintroduction programme in Oman and establishment of the 'World Herd'. Top row displays the years when the animals were captured, translocated or released and the country where the event took place. Middle boxes show the major event and activity at that time. Bottom boxes show the number and sex of animals which had been used [7,16,17]. Additional details can be found at the electronic supplementary material.

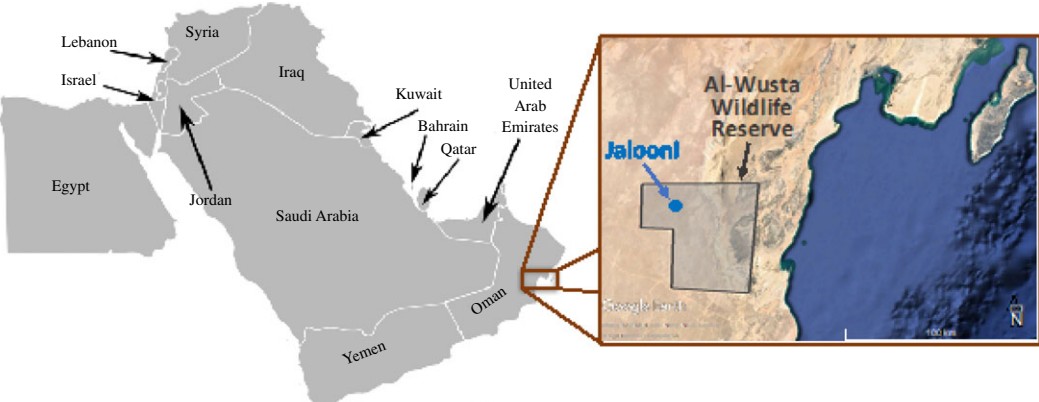

**Figure 2.** Map showing the location of the Al-Wusta Wildlife Reserve (WWR; total area 2824 km$^2$) in the Arabian Peninsula (brown inset). Jalooni is located in the northwest part of the Al-Wusta Wildlife Reserve (blue point; 18°45′ to 22°55′ N and 56°06′ to 57°55′ E). Map of the Arabian Peninsula created by Lokal Profil and licensed under CC-BYSA-2.5.

'World Herd', originated from a small number of founders. This founder effect and a skewed reproductive success in males have probably increased levels of inbreeding in the population [19].

Random mating, as currently performed at WWR, is a type of management that can maintain adequate levels of genetic diversity [20]. However, the acquisition of genetic data could allow the transition to a strategy based on the management of individuals or groups to maximize genetic diversity. This opportunity has been identified by the WWR management and consequently taken action to evaluate the current genetic variation and the genetic contribution of the different population sources to maximize the genetic diversity of the Arabian oryx. To address this, here, we investigated mtDNA CR sequences and whole genome Restriction site-associated DNA sequencing (RAD-sequencing) single nucleotide polymorphisms (SNPs) from 138 Arabian oryx individuals held at the WWR in Oman and 36 historical samples.

# 2. Material and methods

## 2.1. Study area and sampling

Blood samples were obtained from 138 randomly selected individuals at WWR (figure 2; Animal Welfare Committee at the Animal Health Research Centre of Oman, no. 392/2014/ CITES Permit Number 172/

2015). For the purpose of this analysis, samples were assigned to one of three groups according to their primary source population (WWR-Oman and WWR-UAE) or whether they were the result of crossing these two groups (WWR-Mix; table 1). Individuals were identified by ear tags, which include the source group. In addition, 36 historical DNA samples representing some offspring of the founders used for the 'World Herd' were included (table 1) for the purpose of comparison [7]. DNA was extracted using the DNeasy Blood kit (QIAGEN, Germany). For the mtDNA analyses, 138 contemporary WWR and 36 historical samples were used (table 1). For RAD-sequencing, we kept the samples with high DNA quality only, which resulted in the sequencing of 115 samples from the WWR and 16 'World Herd' historical samples.

## 2.2. PCR and sequencing of mtDNA

mtDNA CR sequences (638 base pairs) were amplified using the primers and PCR conditions described in El Alqamy et al. [11], and PCR products were sequenced using an ABI-310 analyser (Applied Biosystems).

## 2.3. ddRAD-sequencing library preparation and sequencing

Double digest restriction-site associated DNA (ddRADseq) libraries were generated using a modified protocol [21] with restriction enzymes SphI and SbfI (New England Biolabs, USA) and a 320–590 bp gel excision, which is described in Bourgeois et al. [22]. In short, DNA quality was assessed via agarose gel electrophoresis on a 1% gel, and only non-degraded DNA (as judged by a tight high molecular weight band against a lambda standard) was selected for the library preparation stage. DNA was quantified using a Qubit Broad Range dsDNA Assay (Thermo Fisher Scientific) and normalized to 7 ng μl$^{-1}$. Each sample was processed in triplicate to enhance evenness of coverage of samples within the library. Individual genomic DNA was restriction-digested using both SbfI and SphI enzymes, and Illumina-specific sequencing adaptors (P1 and P2) were then ligated to fragment ends. The pooled samples were size selected (320–590 bp fragments) by gel electrophoresis and PCR amplified (15 cycles), and the resultant amplicons (ddRAD library) were purified and quantified. Combinatorial inline barcodes (five or seven bases long), included in the P1 and P2 adaptors, allowed each sample replicate to be identified post-sequencing. The ddRAD library was sequenced on the Illumina MiSeq Platform (a single paired-end run; v. 2 chemistry, 2 × 160 bases). Positive control samples were run so that comparisons could be made between all libraries.

## 2.4. mtDNA analyses

Consensus mtDNA CR sequences were obtained for each individual and aligned using the global alignment algorithm of free-end gaps at a cost matrix of 93% similarity [23]. Variable sites and haplotypes from the aligned sequences were identified using the online fasta sequence toolbox Fabox [24]. These sequences were compared with 12 mtDNA CR haplotype sequences from GenBank referred to in this paper as reference/published sequences [12,25]. For consistency, in the current study, we followed the nomenclature proposed earlier for labelling mtDNA haplotypes viz. A to L [11].

The genetic diversity and differentiation of the WWR and historical samples were analysed in this study using DnaSP v. 5.1 [26]. This included the number of haplotypes ($h$), the haplotype diversity ($Hd$), the average number of nucleotide differences ($K$) and the nucleotide diversity ($\pi$) using the model described by Nei [27]. Relationships between haplotypes were assessed with the program PopART [28] using a median-joining network (MJN) and the program's default parameters. The MJN approach was chosen as it is suitable for intraspecific analysis as previously recommended [29–31].

## 2.5. ddRAD-sequencing analyses

### 2.5.1. Variant calling and filtering

Raw FASTQ files from the MiSeq runs were demultiplexed into unique reads for each individual sample using the process_radtags command in STACKS v. 1.09 [32]. The reads were shortened to 140 bp to obtain equal length sequences and then filtered for overall quality. To improve the quality of our dataset and reduce genotyping errors that might have arisen during library preparation and SNP calling [33], we discarded individuals with greater than 50% of missing data. We also discarded loci that were monomorphic, were not present in all groups, had greater than 10% of missing data and had a minor

**Table 1.** Details of samples from the historical 'World Herd' and the current WWR herd used for this study.

| group | reserve/zoo | samples used for mtDNA analyses | samples used for ddRAD analyses | description |
|---|---|---|---|---|
| UK founder[a] | London | 1 | 0 | a male born in London RP (London Zoo) which originally came from 'World Herd' and translocated to KSA in 1989 |
| KSA founder[a] | Thumamah | 5 | 0 | these were translocated from the 'World Herd' in USA to KSA |
| USA founder[a] | SD-WAP | 4 | 1 | San Diego Wildlife Safari Park, Escondido, USA |
| | San Diego | 1 | 0 | San Diego Zoo, San Diego, USA |
| | Phoenix | 2 | 0 | Phoenix Zoo, Phoenix, Arizona, USA |
| | Brownsville | 1 | 0 | Gladys Porter Zoo, Brownsville, USA |
| | Los Angeles | 1 | 1 | Arabian oryx 'World Herd' Los Angeles zoo and botanical gardens |
| Oman founder[a] | Jalooni | 18 | 10 | Jalooni Reserve Office for Conservation Advisor, Muscat, Oman, Currently Office for Conservation of the Environment. These samples represented the founder of Omani Arabian oryx reintroduction project in 1980s. |
| UAE founder[a] | Al-Ain zoo | 3 | 0 | Al-Ain Zoo, Al-Ain, UAE. These animals represent the founders of reintroduced oryx of Al-Ain Zoo in UAE. |
| WWR-Mix | WWR-Mix | 108 | 88 | WWR-Mix represent the offspring which results from crossbreeding of Oman-Herd and WWR-UAE which were translocated to the WWR in late 2011. |
| WWR-Oman | WWR-Oman | 13 | 10 | WWR-Oman Herd: represent oryx which were retrieved from the wild during 1996–2007 |
| WWR-UAE | WWR-UAE | 17 | 15 | WWR-UAE Herd: represent the Arabian oryx introduced to WWR in 2011 |
| | Total[b] | 174 | 125 | |

UK, United Kingdom; KSA, Kingdom of Saudi Arabia; UAE, United Arab Emirates; WWR, Al-Wusta Wildlife Reserve.

[a]Samples obtained from Marshall et al. [7].

[b]The difference in the number of samples between those used for mtDNA and RAD-seq was due to the suitability and quality.

Further information for these samples can be found in electronic supplementary material, table S3.

allele count (MAC) of less than 3. We also discarded loci with significant departure from Hardy–Weinberg proportions, within any one group after Bonferroni correction with a *p*-value of less than 0.05. To calculate deviation from Hardy–Weinberg proportions, we assumed five groups: USA founder, Oman founder, WWR-Mix, WWR-Oman and WWR-UAE (see §1 for a description of these groups). Additionally, we discarded loci with sequences that are too similar which might be possible paralogues (a pair of genes that derives from the same ancestral gene and now reside at different locations within the same genome) by using the function gl.filter.hamming from the R package dartR and a threshold of less than 10 base pairs of difference between sequences.

### 2.5.2. Inference of ancestry and patterns of genetic variation analyses

To infer the genetic ancestry proportion of each individual, we used a Bayesian clustering model implemented in the software fastSTRUCTURE [34]. The algorithm of fastSTRUCTURE approximates the inference of the admixture model of the software STRUCTURE [35]. The admixture model assumes that each individual has a certain proportion of ancestry from one or more of distinct populations or sources ($K$). The model's Bayesian algorithm, in essence, attempts to find the number of populations or sources ($K$) at which population genetics parameters (i.e. Hardy–Weinberg equilibrium within populations and linkage equilibrium between loci) are maximized.

In fastSTRUCTURE, we modelled the number of populations ($K$) from one to eight, with 20 replications for each. We then used a pipeline framework developed by Clumpak [36]. This software automates the post-processes of aligning the results of different values of $K$ using the software Clumpp [37] and the graphical display of the results using the software Distruct [38]. To determine the number of distinct populations that best explains the genetic structure of the dataset, we used a utility provided by fastSTRUCTURE (i.e. chooseK). Additionally, to identify the number of populations that best approximates the marginal likelihood of the data, we extracted and plotted the marginal likelihood of each run of $K$ and averaged it across the 20 replications.

To visualize the spatial pattern of genetic variation present in the genotyped individuals, we performed principal component analysis (PCA) with the help of the R package dartR [39]. PCA is a technique that does not rely on any genetic assumption to identify the main components that explain the structure within the data. The technique uses the top principal components (PCs), which retain the greatest variance in the data, to project the samples onto a series of orthogonal axes [40].

### 2.5.3. Genetic variation within and between groups

To measure genetic variation within groups we used the *q*-profile [41], a spectrum of measures, whose contrasting properties provide a rich summary of diversity, including allelic richness ($q = 0$), Shannon information ($q = 1$) and heterozygosity ($q = 2$). Allelic richness was standardized to be used among groups of unequal sample sizes using the technique of rarefaction [42] implemented in the R package Hierfstat [43]. These measures are converted to a common scale of effective numbers (Hill's numbers) to allow a direct comparison between these measures. We also calculated the number of private alleles (i.e. alleles not present in other populations), the Shannon index [44], observed heterozygosity, expected heterozygosity and inbreeding coefficient in each group. As a measure of genetic variation between populations (i.e. genetic differentiation), we used $F_{ST}$ following Weir and Cockerham [45], and CIs and *p*-values were generated by bootstrapping across loci. All the above statistics were calculated using the R packages: dartR, Poppr [46] and Hierfstat. To identify the groupings contributing the most to the variance of genetic variation, we performed analyses of molecular variance (AMOVA; [47]) as implemented in the R package Poppr using 9999 permutations.

The effective population size (*Ne*) is related to genetic drift and thus predicts rates of loss of neutral genetic variation, fixation of deleterious and favourable alleles and the increase in inbreeding experienced by a population [48]. *Ne* was estimated using the linkage disequilibrium method of Waples and Do [49] as implemented in the software *Ne* Estimator v. 2.1 [50].

### 2.5.4. Identification of potential mating pairs to prevent inbreeding

To identify potential pairs of individuals whose crossing might prevent inbreeding, we used the mean probability of identity of descent across all loci that would result from all the possible crosses of the individuals that were sampled. Two or more alleles are identical by descent (IBD) if they are identical copies of the same ancestral allele in a base population. IBD was calculated by an additive

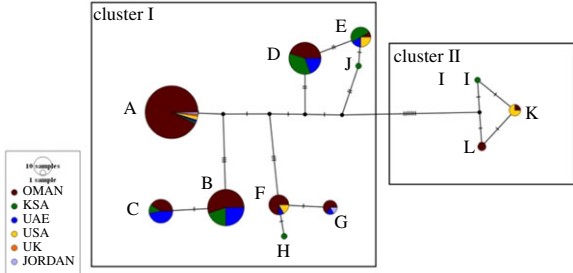

**Figure 3.** Median-joining network (MJN) showing the relationships and clustering of 12 mtDNA CR haplotypes (A to L) including those from published studies. Circle sizes are proportional to the haplotype frequencies, and the length of the lines corresponds to the number of mutations connecting the haplotypes except for the branch leading from the median vectors (black small circles) that joins clusters I and II, which represent seven mutations. Horizontal small lines on branches represent the mutational steps between haplotypes. Further details on the distribution of the mtDNA haplotypes found among the other captive and reintroduction programmes in the Arabian Peninsula are provided in electronic supplementary material, table S1.

relationship matrix approach developed by Endelman [51] as implemented in the R packages dartR and rrBLUP. This additive relationship matrix is a theoretical framework for estimating a relationship matrix that is consistent with an approach to estimate the probability that the alleles at a random locus are identical in state.

We constructed a pedigree by identifying parents and siblings using the R package Sequoia [52] and plotted the pedigree results using the package kinship2 [53]. To identify inbred individuals (i.e. offspring whose parents were close relatives), we estimated inbreeding coefficients for each individual using two different statistics as described in Keller *et al.* [54]: (i) $F_{alt}$, where homozygous loci are weighted with the inverse of their allele frequency using the software GCTA [55] and the command –ibc, and (ii) $F_h$, which is a deviation in homozygosity from its Hardy–Weinberg expectation using the software PLINK [56] using the –het command.

# 3. Results

Relatedness analyses allowed us to identify that four of the samples (410, 454, 471 and 472) were assigned to the wrong groups within the WWR, possibly due to mistakes in the tagging of individuals or during sampling (electronic supplementary material, figure S7). After confirming with WWR management, these samples were assigned to the correct groups for all the downstream analyses.

## 3.1. mtDNA haplotypes and clustering

mtDNA control region sequences of 638 base pairs in length were generated for all 174 oryx samples studied here. Two WWR samples were excluded due to the low-quality reads. A total of 24 polymorphic sites were found across the sequences of the Arabian oryx population in Oman which resulted in nine haplotypes (seven at the WWR and two in historical samples; electronic supplementary material, figures S1–S3). Haplotype sequences are available from GenBank (accession numbers KX696999 to KX697007). Among the seven haplotypes (A–G) found in the WWR groups, haplotype A was the most frequent (49%). Haplotype E had the lowest frequency (less than 1%) and was present in a single individual of the WWR-Mix group. This haplotype was also present in the KSA, USA and UAE groups.

The other five haplotypes (B, C, D, F and G) were shared by three or more of the WWR individuals and were present in 2% to 20% of the samples. Molecular diversity indexes of the WWR individuals were higher in the WWR-Mix group ($n = 15$; $\pi = 0.005$; $K = 3.81$) than in the WWR-Oman ($n = 12$; $\pi = 0.004$; $K = 3.17$) and WWR-UAE ($n = 8$; $\pi = 0.002$; $K = 1.34$) groups, but still lower than those observed among historical samples ($n = 24$; $\pi = 0.009$; $K = 6.85$). Compared with other Arabian oryx captive programmes in the Arabian Peninsula and historical samples (electronic supplementary material, table S2), the WWR as a whole contains 58% of the extant mtDNA haplotype diversity.

MJN analysis (figure 3) did not show a specific clustering pattern, but interestingly, it displayed seven mutations that splits into two groups of haplotypes, named cluster I (haplotypes A–H and J) and cluster II (haplotypes K, L and I). Within cluster I (figure 3), three subclusters (A/B/C, G/F/H and D/E/J)

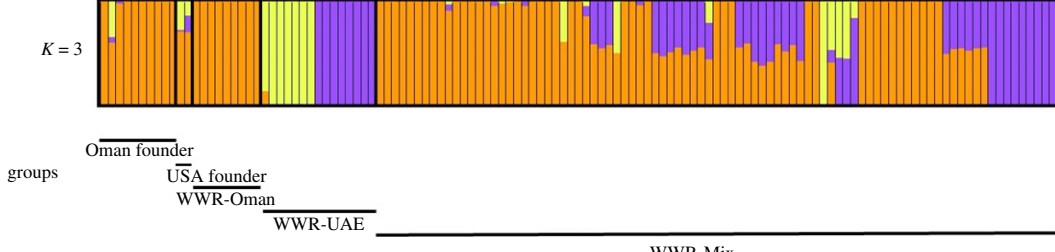

**Figure 4.** Bayesian clustering analysis based on 1091 loci of the five Arabian oryx groups (Oman founder, USA founder, WWR-Oman, WWR-UAE and WWR-Mix) using the software fastSTRUCTURE [33]. Individuals are shown as vertical bars coloured in proportion to their estimated ancestry within each of the inferred populations ($K = 3$).

appear to be formed. WWR individuals shared at least three haplotypes with historical Omani samples. Haplotype G, identified in Omani historical samples, was shared with individuals from the WWR, UAE and Jordan. Individuals from the USA captive programme, from which some of the individuals of the WWR groups were sourced, showed four haplotypes: A, E, F and K. Haplotypes E, H and K observed in the USA captive programme were also observed in the WWR individuals.

## 3.2. ddRAD-sequencing and genotyping

An average of 295 000 paired-end reads of 82–86 bp per sample and in total approximately 82 million reads were produced after barcode trimming, cleaning and quality checking. Five individuals that had more than 50% of missing data were not included in the analyses. Initially 3936 loci were genotyped in 135 individuals. As a result of the filtering, 38 monomorphic loci, 1343 loci not present in all groups, 640 loci that had more than 10% of missing data, 824 loci that had a MAC of less than 3, 0 loci that had a significance departure from HW and 0 loci that had similar sequences were excluded. Our final dataset comprised 1091 loci from 125 individuals. These data are deposited at the National Center for Biotechnology Information, BioSample database of the sequence read archive with the accession nos. SAMN17776257–SAMN17776381.

## 3.3. Inference of ancestry

The inferred number of populations ($K$) that best explained the genetic structure of the dataset according to tool provided by fastStructure (i.e. chooseK) were between 2 and 6. However, at close inspection of the marginal likelihood of each run of $K$, we observed that $K = 3$ had the greatest marginal likelihood, suggesting that the genetic structure of the dataset is best explained by the presence of three different genetic population ancestries (figure 4 and electronic supplementary material, figure S4). Admixture of WWR-Oman and WWR-UAE in the WWR-Mix group was observed, which was expected given that the individuals of the latter group are the offspring of the two former groups. The WWR-UAE group showed two ancestry or distinct population sources.

PCA (figure 5) indicates the presence of three distinct populations. PC1 explained 11.1% of the variance between samples and separated the WWR-UAE samples from the other groups. UAE samples clustered into two distinct groups. PC2 explained 4.9% of the variance in the data and separated the WWR-UAE individuals from the WWR-Mix individuals (figure 5). These groups are a mixture of the founders and descendants of the 'World Herd', two major sources of the WWR programme (WWR-Oman and WWE-UAE) and their offspring (WWR-Mix). One group consisted of USA and Oman founders, WWR-Oman and WWR-Mix individuals, while the other two groups consisted of both WWR and WWR-UAE individuals.

## 3.4. Genetic variation within and between groups

All the groups of the Arabian oryx studied here show similar levels of heterozygosity and low levels of inbreeding except for the WWR-UAE ($n = 15$), which shows a higher inbreeding as measured by the statistics $F_{IS}$, $F_h$ and $F_{alt}$ (table 2 and electronic supplementary material, table S7). Measurements of genetic variation within groups ($q$-profiles; figure 6) show a steep decline in $q = 0$ compared with $q = 1$ and $q = 2$, with the exception of the USA founder group ($n = 2$). Measurements of private alleles

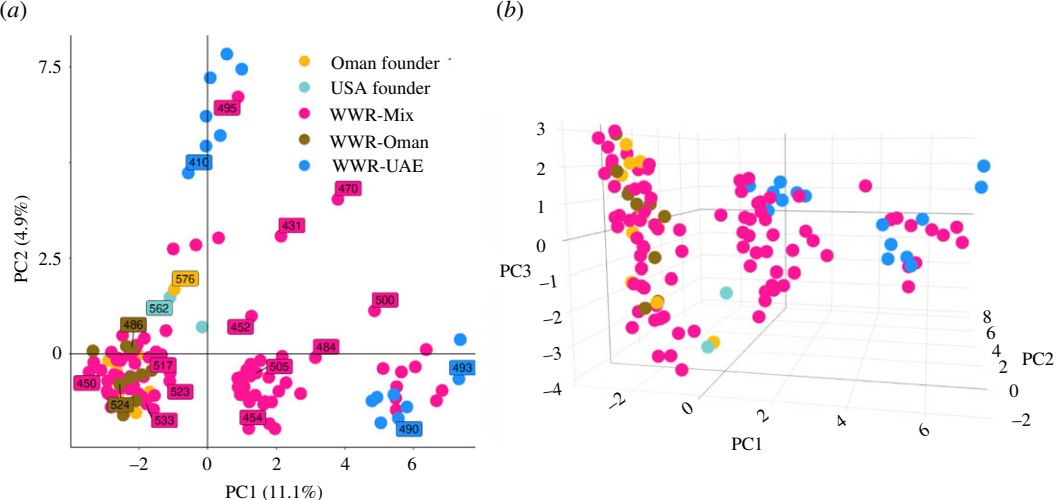

**Figure 5.** Principal component analysis (PCA) using the first two principal components (*a*) and the first three principal components (*b*) that explain the variance in the dataset to project the samples onto orthogonal axes. At visual inspection, it can be observed the presence of three main clusters. Labels in (*a*) show the identification number of some individuals for reference purposes.

indicate that most of these, even those from the founder groups, are present in the WWR-Mix group (electronic supplementary material, table S5). AMOVA analyses and $F_{ST}$ values were significantly different from zero (table 3; electronic supplementary material, table S6) indicating that the main source of variation is mostly within groups rather than between groups.

## 3.5. Parentage analysis and inbreeding estimation

By attempting the reconstruction of the pedigree of the WWR individuals, 10 familial relationships were inferred: seven pairs of full sibling relationships (electronic supplementary material, figure S5) and three pairs of parent–offspring relationships (electronic supplementary material, table S8).

From the three contemporary groups (WWR-Mix, WWR-Oman and WWR-UAE), WWR-Oman is the group that was the least inbred (approximately 25% individuals had positive inbreeding coefficients; electronic supplementary material, table S7) and had the largest effective population size ($Ne = 44$; table 2). By contrast, WWR-UAE was the most inbred group (greater than 75% individuals have positive inbreeding coefficients) and had the lowest effective population size ($Ne = 2$). Electronic supplementary material, table S9 presents the inbreeding estimates of all the individuals analysed, ordered from the most inbred individual to the least inbred.

IBD results (figure 7; electronic supplementary material, figure S6) obtained by the additive relationship matrix agreed with the results of the PCA and fastSTRUCTURE results in identifying three clusters, each including related individuals. In figure 7, two clusters are formed by the WWR-UAE individuals and one formed by the WWR-Oman individuals, while in electronic supplementary material, figure S6, individuals from WWR-Mix are spread between these three clusters formed by WWR-UAE and WWR-Oman groups.

# 4. Discussion

## 4.1. Multiple maternal contributions to the genetic make-up of WWR captive programme

Shared haplotypes (A, F and E) between the WWR individuals and the historical samples from the USA captive breeding programmes suggest that the 'World Herd' has been a major contributor to the WWR programme. The additional four haplotypes identified in the WWR individuals were sourced from the Arabian Peninsula, including KSA and UAE [11,12,15]. For instance, haplotypes C, D and B identified in the WWR-UAE group suggest that UAE is the country of origin. However, it is uncertain whether the UAE population, which was translocated to WWR in 2009, originated from the 'World Herd' or from a private collection in Abu Dhabi or from both [57]. Separation of the haplotypes into two major clusters I and II, which were separated by seven mutations, might be an indication that there was an

**Table 2.** Summary statistics calculated in each group. Standard deviation is shown in parentheses. $H_O$, observed heterozygosity; $H_e$, expected heterozygosity; FIS, inbreeding coefficient; $N_e$, effective population size. $N_e$ in the USA founder group was not calculated due to the low number of individuals sampled ($n = 2$).

| group | sample size | allelic richness | Shannon index | Ho | He | FIS | Ne | Ne CI low–CI high |
|---|---|---|---|---|---|---|---|---|
| Oman founder | 10 | 1.268 (0.194) | 0.386 (0.255) | 0.263 (0.223) | 0.268 (0.194) | 0.023 (0.312) | 16 | 15.2–16.8 |
| USA founder | 2 | 1.259 (0.281) | 0.282 (0.301) | 0.257 (0.325) | 0.259 (0.278) | 0.007 (0.371) | NA | NA |
| WWR-Mix | 89 | 1.281 (0.166) | 0.431 (0.211) | 0.274 (0.170) | 0.281 (0.166) | 0.022 (0.136) | 22.4 | 22.2–22.7 |
| WWR-Oman | 9 | 1.270 (0.195) | 0.387 (0.255) | 0.281 (0.235) | 0.270 (0.195) | −0.027 (0.288) | 44.3 | 39.4–50.4 |
| WWR-UAE | 15 | 1.256 (0.193) | 0.374 (0.261) | 0.233 (0.199) | 0.256 (0.193) | 0.072 (0.292) | 2.1 | 2–2.1 |

The lower the inbreeding coefficient values, the lower the level of inbreeding.

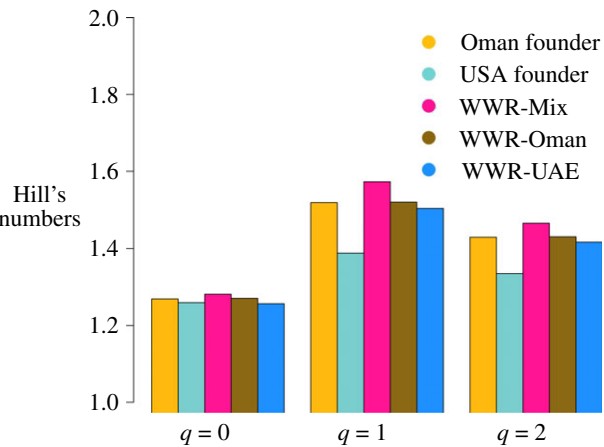

**Figure 6.** The $q$-profile [40] showing Hill's numbers: allelic richness ($q = 0$), Shannon information [43], ($q = 1$) and expected heterozygosity ($q = 2$) converted to a common scale of effective numbers (Hill's numbers).

**Table 3.** Pairwise genetic differentiation ($F_{ST}$) following Weir and Cockerham [45] CIs and $p$-values were generated by bootstrapping across loci using the R package dartR [37].

| group 1 | group 2 | $F_{st}$ | lower bound CI limit (95%) | upper bound CI limit (95%) | $p$-value |
|---|---|---|---|---|---|
| USA founder | Oman founder | 0.082 | 0.063 | 0.100 | <0.01 |
| USA founder | WWR-Mix | 0.063 | 0.047 | 0.077 | <0.01 |
| USA founder | WWR-Oman | 0.091 | 0.067 | 0.108 | <0.01 |
| USA founder | WWR-UAE | 0.121 | 0.098 | 0.143 | <0.01 |
| Oman founder | WWR-Mix | 0.027 | 0.023 | 0.034 | <0.01 |
| Oman founder | WWR-Oman | 0.008 | 0.001 | 0.013 | <0.01 |
| Oman founder | WWR UAE | 0.121 | 0.109 | 0.133 | <0.01 |
| WWR-Mix | WWR-Oman | 0.010 | 0.005 | 0.013 | <0.01 |
| WWR-Mix | WWR-UAE | 0.054 | 0.048 | 0.060 | <0.01 |
| WWR-Oman | WWR-UAE | 0.113 | 0.100 | 0.128 | <0.01 |

appreciable level of divergence within the species long before it became extinct from the wild in 1972. Based on the current data, we surmised that cluster I may have been widely distributed in the Arabian Peninsula, while cluster II could have been restricted to the Oman and the KSA region. However, further historical samples would be needed to reach a more definitive conclusion.

## 4.2. Three genetic ancestries and moderate levels of genetic diversity

Here, we describe relatively similar genetic diversity among the founders of the 'World Herd', the population sources of the WWR programme (WWR-Oman and WWR-UAE) and the offspring of these sources (WWR-Mix). Our results reveal that the genetic composition of the Arabian oryx individuals at the WWR is best explained by the presence of three different genetic groups, two represented by UAE individuals and the other from the founders of the 'World Herd'. The two different ancestry sources from the UAE individuals are consistent with microsatellite data (six loci) that the Arabian oryx population in UAE have different ancestors from those of the 'World Herd' [7]. Furthermore, a more recent microsatellite study (13 loci; 61 individuals) of the UAE captive programmes, whose individuals were sourced from three private and zoo collections, showed some level of genetic differentiation between the two major populations sources [11]. This has been attributed to high genetic drift caused by using a small number of founders and isolation across multiple generations between the two main founder groups of the breeding programmes in UAE [11]. Our SNP study shows slightly lower levels of heterozygosity (0.258) for the individuals sourced from

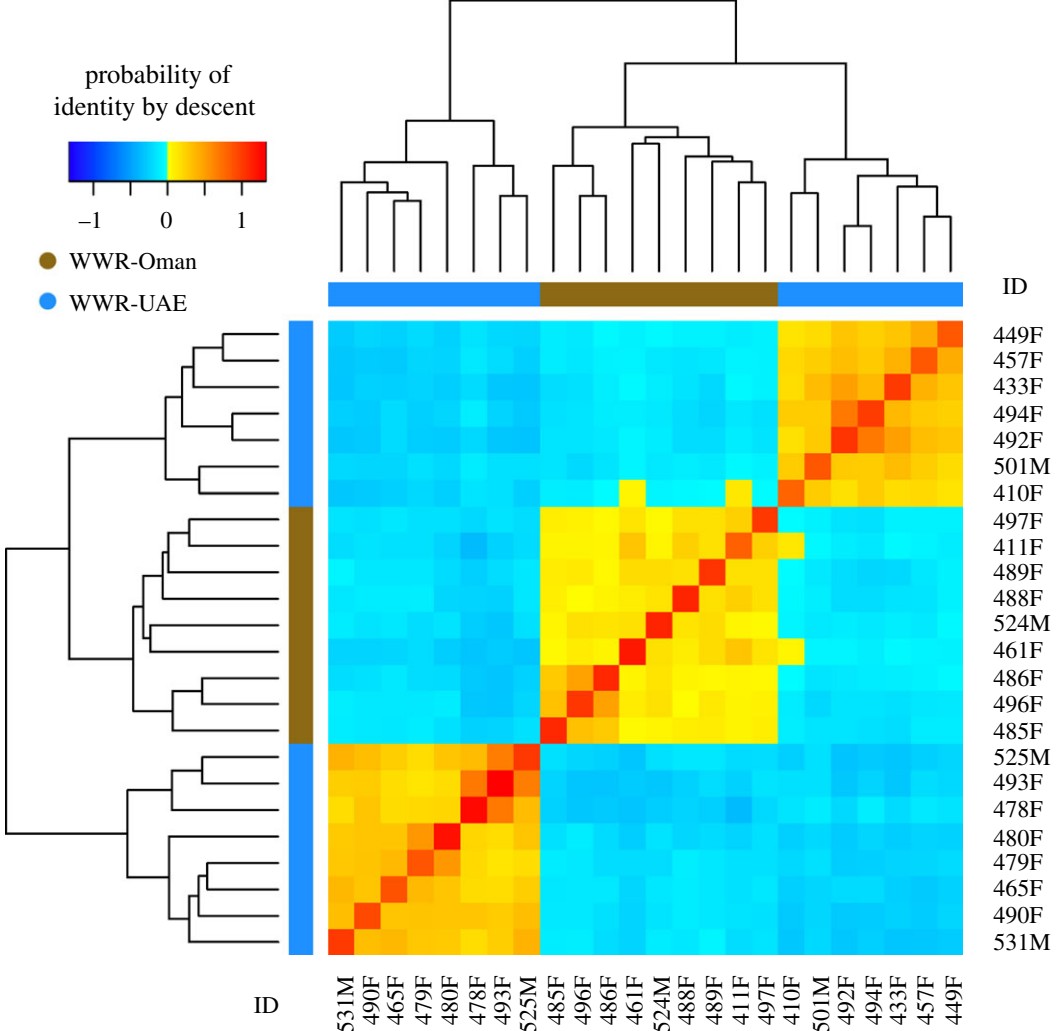

**Figure 7.** Heatmap of the probabilities of identity by descent in the group formed by the herds of WWR-Oman and WWR-UAE. As described by Endelman & Jannink [67], each diagonal element has a mean that equals $1 + f$, where $f$ is the inbreeding coefficient (i.e. the probability that the two alleles at a randomly chosen locus are IBD from the base population). As this probability lies between 0 and 1, the diagonal elements range from 1 to 2. Because the inbreeding coefficients are expressed relative to the current population, the mean of the off-diagonal elements is $-(1 + f)/n$, where n is the number of loci. Yellow and red colours indicate those individuals that are more related to each other. The identification number of each individual is shown in the margins of the figure, where the last letter denotes whether the individual is male (M) or female (F).

the UAE compared with the other WWR groups (0.270–0.282) and the founder groups (USA founder and Oman founder) in our study (0.263–0.290). This aligns with the reduced genetic diversity found in the UAE individuals described by microsatellites [11]. Our study also revealed considerable genetic admixture for the WWR-Mix group, which was expected. Additionally, we observed that the three genetic ancestries that were identified are randomly distributed among the three management herds (enclosures) at WWR (electronic supplementary material, figure S8). This observation indicates that genetic patterns presented in this study were not due to drift based on isolation of these three herds. We expect that this random pattern persists under the current management, which is based on random rotation of individuals between enclosures.

Genetic diversity of the WWR-Mix group shows a slight increase compared with the historical/ founder individuals. Our analyses showed slightly lower levels of *He* in comparison with *Ho* in all groups analysed except for the WWR-Oman group. It is important to note the challenges of comparing results obtained through SNP-based data (as in this study) with that from microsatellites (most of the previous studies). For instance, and in general, heterozygosity as measured by SNPs (i.e. based on biallelic loci) tends to be lower than heterozygosity measured by microsatellites (which usually display more than two alleles). Furthermore, SNP data provide a more accurate representation

of the genome, because current sequencing technologies allow for genotyping thousands of markers compared with tens of markers for typical microsatellites datasets. After acknowledging the above caveats, our results showed the same trend as in microsatellite-based studies [6,7,11,12]. Specifically, Marshall *et al.* [7] reported low population differentiation among four captive programmes in the Arabian Peninsula and the 'World Herd', based on 343 individuals including 77 samples from the Arabian Oryx Sanctuary (AOS). Consistent with our SNP study, microsatellites showed substantial genetic admixture among these populations, which is attributed to the translocations and exchange of individuals between programmes [7]. For the AOS individuals in this study—and it is unclear whether individuals were sourced from the founders of the WWR-Oman group—microsatellites showed a slightly lower level of heterozygosity ($He = 0.470$) compared with individuals from other programmes ($He = 0.509$–$0.622$; 7). Interestingly, our SNP results showed that there was a slight increase of heterozygosity in the WWR-Mix group compared with the parental sources (WWR-Oman and WWR-UAE) and the founders of the 'World Herd'. This result is encouraging as it shows that a management based on random mating can conserve genetic diversity at adequate levels.

The use of the $q$-profile in this study provides additional insights into the genetic composition and variation of the WWR groups. For instance, $q = 0$ is sensitive to rare alleles (i.e. alleles present in low frequencies in the population), but its estimates are highly dependent on sampling size, whereas $q = 2$ shows the opposite trend [41]. In the absence of an evolutionary process that is strongly influencing genetic variation, such as selection, dispersal or genetic drift, we expect to observe similar values of the Hill's numbers of the three different $q$-values, that is, we expect to observe an evenness of the allelic distributions across the three $q$-values. What we observed in the WWR is a decline in $q = 0$ compared with $q = 1$ and $q = 2$ which could be an indication that genetic variation, especially rare alleles, has been lost and is not distributed evenly across each group but harboured within individuals. This observation is probably a reflection of the species' history of frequent founder effects which generally result in a depletion of genetic diversity. Together, these results suggest that the main source of variation is mostly within groups and not between groups. Furthermore, we observe that most alleles, including private alleles from the founder groups, are present in the WWR-Mix. Our results demonstrate the advantage of using the $q$-profile. For instance, we observe that genetic variation measured by heterozygosity is relatively uniform between groups, except in the USA founder group. This effect is probably because heterozygosity reflects the variation of the most frequent alleles, which is unaffected even after population bottlenecks [58]. The $q$-profile also revealed that the WWR-Mix group harbours more genetic variation than the other groups. This has been a fortunate outcome given the random mating approach and absence of genetic understanding of the founders used at WWR to date. This study generated baseline information regarding several measures of genetic diversity and relatedness (e.g. figures 3 and 7 and table 2), which will be beneficial for the future management at WWR. For instance, this information can be used to guide the breeding programme including setting defined goals, such as reaching a certain effective population size and specific levels of genetic variation and inbreeding.

## 4.3. Implications for management and conservation at WWR

Understanding the genetic structure of the contemporary populations and their connection with historical populations is crucial to ensure the restoration of the Arabian oryx into their historical distribution range and to guide the management of the captive populations as demonstrated in other species [59–62]. With the support of genetic data, such as the data generated in this study, the breeding programme at the WWR could move from a random breeding approach to one with a clear strategy aimed at maximizing genetic diversity and evolutionary potential and minimizing the effects of inbreeding [63]. However, it is necessary to recognize the feasibility of implementing different types of management strategies, as well as their trade-offs and genetic implications, depending on the availability of physical, human and financial resources. Possible types of management might range from random mating, as currently implemented at WWR, to more intensive strategies such as the selection of groups or even specific individuals for breeding.

Our results demonstrated that strategies based on random mating could be reasonably successful at maintaining genetic diversity and reducing inbreeding in some herds. Conversely, our results also showed that the three ancestral groups identified are not represented evenly across the three herds under management at WWR, which might eventually result in the loss of rare alleles.

A group management strategy might involve genetic monitoring for genetic diversity and rotation of groups of individuals between the three enclosures. Approaches like this are currently being carried out

for the threatened sable antelope (*Hippotragus niger*) in the USA (Source Population Alliance— Conservation Centers for Species Survival; [64]). More specifically, this group strategy might involve the establishment of satellite breeding groups, which would include a smaller breeding nucleus with multiple females, complemented with a seasonal rotational male breeding scheme, so that females have the opportunity to breed with other representative males [63] from the genetic lineages identified in this study. Given that genotypic data is available for only approximately 25% of the WWR population, it is important to maximize this genetic knowledge to select males and females representing the different genetic lineages found at WWR and complement this with studbook information for those animals for which information is not available. The advantage of this approach is that it does not require the regular handling of individuals, and the genotyping of a small proportion of individuals is often sufficient to guide this type of management.

The most intensive strategy would be based on breeding of specific individuals. For instance, this strategy could involve the reproduction of individuals carrying low frequency haplotypes such as Haplotype E, which was identified in a single individual at WWR. For this strategy, some of our results such as heatmaps of identity by descent (figure 7 and electronic supplementary material, figure S6), pedigree reconstruction and inbreeding estimates (electronic supplementary material, table S7) could be used to select the most suitable pairs of individuals for breeding or individuals to be released into the wild. The implementation of this individual-based strategy could also involve the use of specialized software to assist with the management of breeding such as PMx [65]. A disadvantage of this strategy is that it requires more specialized infrastructure, regular handling of individuals and generating a more complete genetic profile of the populations under management.

A further measure that could be advantageous for the conservation of Arabian oryx is the development of a cryopreservation programme of genetic material including semen, ova, cell-lines or embryos [66]. This programme could act as an insurance resource for the long-term preservation of parental and most representative ancestries. The choosing of individuals for this programme could be based on genetic ancestries that have been identified across the different programmes in the Arabian Peninsula, including those outlined for each WWR group in figure 5.

It is worthwhile to mention that efforts to generate the reference genome of the Arabian Oryx are underway (Dr Brooks 2021, personal communication; University of Florida). This important resource will make it possible to map the SNPs used in this study, which will allow comparisons across experiments, investigators and time. We acknowledge that it is also important to retain samples, especially historical samples, for future analyses at the genome level.

A long-term and global goal for the Arabian oryx conservation community should be to develop a management plan whose aim is not just the increasing of population size but the maximization and retention of genetic diversity over time [17]. We recommend the implementation of a genetic-based management approach for the Arabian oryx at WWR, which could include the monitoring of genetic diversity statistics and biobanking of samples. These measures will certainly minimize the major threats to the survival of this important reintroduction programme. The resources generated in this study provide some useful baseline information to allow WWR management to be able to inform this going forward and generate SNP data for the rest of the animals.

Ethics. Animal Welfare Committee at the Animal Health Research Centre of Oman, no. 392/2014.

Data accessibility. All data available in the GenBank Sequence Read Archive (SRA) accession nos. KX696999 to KX697007.

Data and relevant code for this research work are stored in GitHub: https://github.com/mijangos81/Oryx and have been archived within the Zenodo repository: https://doi.org/10.5281/zenodo.5933824.

Authors' contributions. Q.A.R.: conceptualization, data curation, formal analysis, resources, writing—original draft; J.L.M.: data curation, formal analysis, writing—review and editing; M.S.K.: writing—review and editing; M.A.A.A.: writing— review and editing; M.H.A.: writing—review and editing; J.K.: data curation; H.S.: data curation, writing—review and editing; K.B.: formal analysis, project administration, writing—review and editing; J.G.: conceptualization, data curation, methodology, project administration, resources, supervision, writing—review and editing.

All authors gave final approval for publication and agreed to be held accountable for the work performed therein.

Competing interests. We declare we have no competing interests.

Funding. Funding for our work was provided by Office for Conservation of the Environment (OCE), Diwan of Royal Court, Oman. Oryx blood and DNA samples were exported to the UK under import permit from the Office for Conservation of the Environment Oman (CITES Certificate no. 172/2015) and under import permit from the UK (CITES Certificate no. 540806/01).

Acknowledgements. We are grateful to Zahran Al Abdulsalam and Sultan Al Bulushi for their support during the sample collection. We thank Mr Yasser Al Salami, The DG of OCE, for providing some of the funding support to undertake this project at the University of Sydney. We also thank Hani Al Saadi, Sultan Al Bulushi, Zahran Al Abdulsalam, Waheed Al Fazari and Sami Al Rahbi for their support in sampling and data collection. We also thank Dr Marshall

T.C. for allowing us to access the historical samples. The Ministry of Higher Education in Oman provided PhD scholarship to QAR.

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
