## [Peer Review File · Royal Society Open Science]

Review History

RSOS-210558.R0 (Original submission)

Review form: Reviewer 1

Is the manuscript scientifically sound in its present form?

No

Are the interpretations and conclusions justified by the results?

Yes

Is the language acceptable?

No

Do you have any ethical concerns with this paper?

No

Have you any concerns about statistical analyses in this paper?

Yes

Recommendation?

Major revision is needed (please make suggestions in comments)

Comments to the Author(s)

The paper represents a significant body of research into the population genetic diversity of Arabian oryx in Oman and implications for future conservation breeding programmes. This species, and particularly its conservation history in Oman, are internationally renowned in conservation reintroductions and this topic will certainly be of broad interest to the conservation community.

The study has been well-designed and the analyses are for the most part appropriate, but the description of the methods causes some confusion over what is being analysed, particularly in terms of the sample groupings.

The description of the results was at times cursory and elsewhere inaccurate. If population genetic analysis is going to be employed and included in the manuscript, the authors need to ensure that it is there for a reason, is accurately presented and adequately explained. Examples of issue are provided in Minor corrections below.

I identified six grammatical corrections / improvements in the abstract. On this basis I suggest that the entire manuscript would benefit from careful revision of the English throughout, by a naive speaker. There appear to be native English speakers in the author list who presumably could do this.

While the manuscript needs quite a bit of work to make it suitable for publication, I feel that there is sufficient basis for a publication in RSOS and if the authors are prepared to make the recommended improvements I would be happy to recommend publication.

Minor corrections:

As the references in this journal are simply inserted as numbers, the authors should try to avoid citing just the number when referring to an actual person. For example '... approach developed by (49)...'. In this case, please write the name of the author (with et al. if necessary) and place the reference number in parantheses afterwards, e.g. '... approach developed by Endelman et al. (49)...'

Line 8 Ethics If laboratory analysis was performed outside of Oman please include details of CITES permits or licences

Abstract:

Line 27 '...founders, and private collections...'

Line 29-31 Worth mentioning how oryx came to be in the UAE

Line 35 '...contain...'. Although probably better to use '...display 58% of observed haplotype...'

Line 39 Rephrase. The SNP data did not create anything.

Line 40 '...identification of those individuals for which mating pairs...'

Line 42 Remove 'the'

Line 43 Remove 'the'

Introduction:

Line 49 Rephrase 'a definitive extinction'. Perhaps 'complete extinction'?

Lines 50 & 51 Implies Phoenix zoo is in (on) the Arabian peninsula

Line 66-70 What do the authors mean by 'low' and 'high' levels of genetic diversity? Perhaps insert 'relatively' here, to communicate that different populations are being compared (assuming they are comparable).

Line 93 If known, has there been any mixing of these three herds since they were created?

Line 110 Error in references

Materials and Methods:

Line 113 This implies that some individuals from the original populations (WWR-Oman and WWR-UAE) were not mixed. However at Line 90 the authors suggest that all animals were potentially mixed. Please clarify.

Throughout the results the authors refer to five groups of oryx: Oman founder, USA founder, WWR Mix, WWR Oman and WWR UAE. However it is not clear from Table 1 where the USA founder animals are, as none appear to have been analysed using RAD-seq. This needs clarifying. Please make Table 1 consistent with other Figures and Tables and explain clearly at the start of the Methods, where it states that 3 groups were used (Line 113).

Line 158 What is meant by a 'Bonferroni correction of <math><0.05</math>'? What was the P-value employed after Bonferroni correction? How many populations were assumed etc.

Line 179 '...rely on...'

Line 198 It is not clear how an estimate of N_e would inform how different groups have been affected by drift / inbreeding, unless you assume the three (or five?) groups had identical starting conditions. Please explain.

Results:

Line 235 Please clarify whether this is considered to be 58% of extant diversity, or 58% of total diversity ever observed?

Line 241 Reference error

Line 268 onwards Description of PCA results appears confused. PC2 differentiates WWR-UAE, not WWR-Oman as far as I can tell.

Line 284 F_{st} values. Some of these are actually quite high (>12%) and they vary a lot: <math><1\%</math> to >12%. Have they been tested for significance? More accurate description pairwise F_{ST} data required for Table 3

Line 288 Replace 'parentage' with 'familial'

Line 294 Commentary on the usefulness of a group for conservation should be moved to the Discussion. Also, consider that the UAE, although showing greater inbreeding, may also carry distinct genetic diversity.

Discussion:

General: The finding of a distinct mtDNA group is one of the most interesting findings of the work and is probably worthy of more discussion. Do the authors recommend increasing the frequency of these haplotypes in the Oman population. Are there any historical (phylogeographic) explanations for this division? Does it mirror any other patterns of differentiation in Arabian fauna?

Line 313 Would replace 'confirms' with 'indicates'.

Line 325 How do the three groups in the nuclear data compare to the three management herds in Oman? Could the results be due to drift based on isolation of these three herds? If not, is the observed pattern of diversity expected to persist under current management? This is important and more Discussion is required here

Line 377 Sentence doesn't make sense

Figures and Tables:

Figure 1

The Legend is extremely long. Recommend reducing if possible, although the information here is valuable - maybe include long version in Supp. Mat..

The Legend states that the middle boxes show conservation status, but I cannot see this.

Figure 2

Suggest straightening and thinning the black line. Its only needs to join the edges of the relevant areas, not the middle.

The maps apart o have been copied from elsewhere. Include information on the source of the original maps.

The right hand map contains place names that are too small to read. Suggest either removing or making bigger.

Figure 3

Legend box and right hand cluster box are missing sides

Again, the legend text is too long, suggest reducing and placing extra text in Supp. Mat.

Figure 4

Final sentence of the legend refers to identification numbers of individuals on the x-axis, but these are impossible to read and are not critically important - suggest removing.

Table 1 Please align the sample categories with the five 'groups' used throughout the results.

Supp Table 3 Column headers (Sigma, Percentage and Phi) need more explanation. What does this table show?

Supp Table 4 Explain Fh and Falt in the legend

Supp Table 7 How do you define 'highest'?

Review form: Reviewer 2

Is the manuscript scientifically sound in its present form?

Yes

Are the interpretations and conclusions justified by the results?

Yes

Is the language acceptable?

Yes

Do you have any ethical concerns with this paper?

No

Have you any concerns about statistical analyses in this paper?

Yes

Recommendation?

Accept with minor revision (please list in comments)

Comments to the Author(s)

General comments

Using mitochondrial sequences and SNP genotypes generated from RAD-sequencing, the authors analyzed the genetic diversity, inbreeding, structure, and kinship among Arabian oryx maintained at the Al-Wusta Wildlife Reserve in Oman, where an important captive breeding and reintroduction program was established for this species, and among a set of historical samples collected in the 1980s, when the first captive breeding programs for the species were initiated. The animals at the Al-Wusta Wildlife Reserve originate from different sources. Therefore, understanding the genetic contributions and consequences of these sources are useful for the conservation breeding and reintroduction programs, which have lacked formal genetic management strategies until now. The data and results the authors have generated represent an important advancement in our understanding of the genetics of Arabian oryx captive populations and also represents a nice model of how genomic data can be directly implemented to inform breeding recommendations (in the immediate future) and reintroductions (a more distant goal).

The manuscript is generally well-written and presents entirely novel data and results. The amount of text devoted to the introduction, materials and methods, results, and discussion sections is well balanced. The cited literature is comprehensive and up to date. The mitochondrial and RAD-seq data have been analyzed in a comprehensive and rigorous manner, and the authors have appeared to pay attention to sample size issues and the underlying assumptions of the analytical methods they applied. I was especially intrigued with the use of the q-profile approach developed for evolutionary and population genetic applications by Sherwin et al. 2017. This approach standardizes estimates of population genetic parameters into effective measures that make them easily comparable across different sampled groups. The figures and tables are generally of good quality and easily interpretable. The additional results presented in the supplementary figures and tables provide important details that support the conclusions. I especially like Supplementary Figure 6 (the larger heatmap of IBD among all sampled

individuals). The manuscript suffers from quite a number of grammatical errors that diminish clarity and readability. I provide comments and recommendations regarding these and other issues for the authors to address, which I think would help improve this important manuscript.

Specific comments

1. Lines 47-48: "IUCN red list of threatened species" This is a formal name and therefore the full name should be capitalized: "IUCN Red List of Threatened Species"
2. Lines 58-59: Sentence awkwardly worded: "The current number of the Arabian oryx is estimated ~1,100 in the wild and ~6,000-7,000 in captivity, zoos, reserves and private collections (1)."
Suggested revision: "The current number of the Arabian oryx is estimated to be ~1,100 in the wild and ~6,000-7,000 in captivity, which includes zoos, reserves and private collections (1)."
3. Lines 78-79: "Eventually, through successful management, the Jalooni population increased to about 400 captive and reintroduced individuals into the wild by 1996 (16)." As written, it's not clear whether the authors' mean a total of 400 individuals from both captive and wild populations were present in 1996 or that by 1996, 400 individuals had been reintroduced into the wild. I assume the former was the intent of the sentence. In that case, remove the phrase "into the wild" to clarify the meaning of the sentence.
4. Line 97: "World Herd" is misspelled in this line.
5. Line 113: "...its primary source population..." As this phrase refers to the "samples" (plural), this should be changed to "...their primary source population..."
6. Line 114: Add a 'the' before "result" in this sentence.
7. Line 122: For the sake of readers, specify the length of the mtDNA CR sequence that was amplified. This is indicated as 638 base pairs in the Results section. However, it's useful to mention this here in the Methods section.
8. Lines 122 and 134: "MtDNA CR" and "mtDNA CR" To improve clarity, these should be qualified by the word 'sequences': "MtDNA CR sequences" and "mtDNA CR sequences"
9. Line 131: Specify the method and instrument used to quantify libraries. I assume using a qPCR?
10. Line 152: "...to obtain equal length of sequences..." Delete the "of" in this sentence.
11. Lines 154-155: What to the authors mean by "individuals sampled more than once"? Were some individuals sequenced more than once in the RAD-seq protocol? If so, why and what proportion of the 135 samples used for the RAD-seq analyses?
12. Line 172: "explain" should be changed to "explains" in this sentence.
13. Line 179: "PCA is a technique that does not relay in any genetic assumption..." I assume the authors meant "PCA is a technique that does not rely on any genetic assumption...?"
14. Line 187: "...to be used in samples of unequal sample sizes..." To improve clarity, change to "...to be used among groups of unequal sample sizes..."

15. Lines 189-190: "...to allow a direct comparison." A comparison of what, exactly? Among groups? Populations? Please specify.
16. Line 191: Add a "the" before "Shannon index."
17. Line 192: Add a "a" before "measure."
18. Lines 194-195: "To identify the group contributing with most to the variance of genetic variation..." Suggested revision: "To identify the groupings contributing the most to the variance of genetic variation..."
19. Line 216: Delete "Yang et al." in this sentence.
20. Line 221: "MtDNA control region (638 bp) was generated..." To improve clarity and grammar, change to: "MtDNA control region sequences 638 bp in length were generated..."
21. Line 225: "Haplotypes sequences are available in GenBank..." Suggested revision: "Haplotype sequences are available from GenBank..."
22. Lines 238-239: In Figure 3: these two haplogroups are referred to as Cluster I and Cluster II. For the sake of consistency, the authors should select one of these - haplogroup or cluster.
23. Line 239: I think the authors mean haplogroup (or Cluster) I here.
24. Line 252: Change "no present" to "not present" in this sentence.
25. Line 254: Add "was" before "comprised" in this sentence.
26. Line 255: Add a "the" before "National Center for Biotechnology Information"
27. Line 259: Awkward wording: "The inferred number of populations (K) that explained the best the genetic structure..." Suggested revision: "The inferred number of populations (K) that best explained the genetic structure..."
28. Line 262: Add a comma after "likelihood" in this sentence.
29. Line 272: "founders/descendent." To improve clarity, this should be changed to "founders and descendants"
30. Line 298: Change "less inbred" to "least inbred"
31. Line 300: Remove the comma after "PCA" in this sentence.
32. Lines 314-315: Awkward sentence construction: "The additional four haplotypes were identified in the WWR individuals sourced from the Arabian Peninsula including KSA and UAE..." Suggested revision: "The additional four haplotypes identified in the WWR individuals were sourced from the Arabian Peninsula, including KSA and UAE..."
33. Line 333: Change "recently" to "recent"
34. Line 343: Add a comma after "group" in this sentence.

35. Lines 351-352: Awkward sentence construction: "After, acknowledging the above caveats, overall our results showed..." Suggested revision: "After acknowledging the above caveats, our results showed..."
36. Line 366: it's not clear what the authors mean about $q=0$ having "sampling problems." Do they mean that the estimation $q=0$ is influenced by sample size of populations measured? Please revise to specify the problems associated with the q -profile approach.
37. Add a comma after "Together" in this sentence.
38. Line 382: Change "by" to "in" - "in the USA founder group."
39. Line 386: Change "use" to "used" in this sentence.
40. Lines 396-397: "...aimed to maximize genetic diversity and control the negative effects of inbreeding and evolutionary potential." How is controlling evolutionary potential a negative effect? I think the authors meant to write: "...aimed to maximize genetic diversity and evolutionary potential and control the negative effects of inbreeding."
41. Line 413: Add a comma after "Arabian Peninsula" in this sentence.
42. Line 415: "long" should be changed to "long-term" in this sentence.
43. Line 417: "separating offspring" - the authors need to indicate when such a separation would occur. I assume after weaning? Or when individuals reach sexual maturity. Such a recommendation has important implications and therefore needs to be better specified.
44. Lines 421-424: Awkward sentence construction: "...establishing satellite breeding groups, this could be smaller breeding nucleus with multiple females complemented with a seasonal rotational male breeding scheme so females could had the opportunity to breed with other representatives males (63) from the genetic lineages identified in this study;" Suggested revision: "...establishing satellite breeding groups, which would include a smaller breeding nucleus with multiple females, complemented with a seasonal rotational male breeding scheme so that females have the opportunity to breed with other representative males (63) from the genetic lineages identified in this study;"
45. Lines 429-430: "from a continuing genetic monitoring program" Do the authors mean using the same genetic approaches as reported in their paper? Please specify, as this will provide useful guidance to all managers of Arabian oryx breeding and reintroduction programs.
46. Line 436: Change "samples" to "sample" in this sentence.
47. Figure 1: Add the word "program" after "reintroduction" in the first sentence of the caption. For the middle boxes of the figure - first middle box: "Conducted of "Operation Oryx" should be revised to "Operation Oryx" conducted". In the fourth middle box, text can be changed to "Reintroduction of "Oman-Herd into the wild." In the bottom boxes, "Operation Oryx" in the third box should be capitalized for the sake of consistency. Similarly, "World-herd" can be changed to "World Herd." in the fourth box.
48. Figure 2: Awkward wording: "Jalooni located in west north of Al-Wusta Wildlife Reserve." Suggested revision: "Jalooni is located in northwestern part of the Al-Wusta Wildlife Reserve." Also, please specify what the purple blotches indicate in the map figure on the left. I assume these are the Arabian oryx reserves in Israel, Kuwait, KSA, Oman and UAE?

49. Figure 3: Change the period to a comma before “Jordan and USA (KU985184, Ochoa et al., 2016)” in the caption. Also, add a comma before “with seven haplotypes (A-G)...” In addition, revise the following sentence: “Further details on the distribution of these haplotypes among other the captive and reintroduction programs in the Arabian Peninsula...” as “Further details on the distribution of these haplotypes among the other captive and reintroduction programs in the Arabian Peninsula...”

50. Figure 4: The clustering plot and individual ID numbers at the bottom need to be increased in size to improve readability. In the current version, the IDs are too small to be read.

51. Figure 7 caption: “Endelman & Jannink19” should be changed to “Endelman & Jannink (2012)”

52. Table 1: Are specific years in the 1980s available for the historical samples?

53. Tables 2 and 3: Can the authors test for significant difference in the values (e.g., Shannon index, Fst) among the groups reported in these tables?

Review form: Reviewer 3

Is the manuscript scientifically sound in its present form?

Yes

Are the interpretations and conclusions justified by the results?

Yes

Is the language acceptable?

Yes

Do you have any ethical concerns with this paper?

No

Have you any concerns about statistical analyses in this paper?

No

Recommendation?

Accept with minor revision (please list in comments)

Comments to the Author(s)

See attached review file (Appendix A).

Review form: Reviewer 4

Is the manuscript scientifically sound in its present form?

No

Are the interpretations and conclusions justified by the results?

No

Is the language acceptable?

Yes

Do you have any ethical concerns with this paper?

No

Have you any concerns about statistical analyses in this paper?

Yes

Recommendation?

Major revision is needed (please make suggestions in comments)

Comments to the Author(s)

RSOS-210558

This manuscript presents a very interesting and worthwhile contribution to the literature on a species that was critically endangered, thought to be extinct in the wild, had its population expanded through managed breeding under human care, was released back into the wild and remains a focal species for private collections. The assessments of genetic diversity previously performed were typically for subpopulations of the overall population, lacked resolving power and made only minor contributions to population management regarding retention of the greatest possible extent of the surviving genetic variation of the species, the Arabian oryx, or white oryx, *Oryx leucoryx*. As the authors state (l 88-90), “The breeding program focused on increasing population size, rather than on increasing genetic diversity.” and (l 99-101), “Additionally, the absence of genetic data has limited the ability of this program to move from a strategy of random mating to a genetic-based (or group-based) management.”

To address the information missing from what has become standard population management practice for conservation of large mammal species, the authors obtained a variety of samples of blood, tissues, or DNA extracts from previous studies and conducted two separate studies. One investigated mitochondrial DNA diversity using PCR amplification and DNA sequencing of 174 individuals, including samples from individuals from the early generations of the breeding efforts. The other used reduced representation library sequencing methods (Rad-Seq) to interrogate presumptively homologous portions of the *O. leucoryx* genome from 135 individuals in the later generations of the breeding efforts.

Usable results were obtained for the mitochondrial analyses from more samples than for the Rad-Seq analysis, a not unexpected result as the authors explained (l 119), “we kept the samples with high DNA quality only.”

A mitochondrial DNA network was generated and, using comparison between the ‘historical’ samples (from the early generations of captive breeding when a pedigree for the World Herd was being kept), and the samples from the recent generations of animals bred in the Sultanate of Oman the authors show that this population (WWR) as a whole contains 58% of the global mtDNA 236 haplotype diversity. There are some indications of structure in the mitochondrial DNA haplotype map, but the total branch length of the network is relatively small and the authors identify two clusters of haplotypes that are separated by seven mutations.

The results of the Rad-Seq analysis are presented several ways: as a PCA plot, a Bayesian cluster analysis and with a heat-map of estimated relatedness. These data and analyses of F_{st} combine to suggest that there was modest genetic differentiation in the groups of individuals that have been incorporated into the current managed population in the Sultanate of Oman. Although admixture

has taken place between the combined groups, there is still some structure within the overall Oman population.

Management goals are alluded to, but the paper might have more impact if concrete examples were given.

Disparate founder contributions can have important impacts in analyses like this one when attempting to manage allele loss. It is hard to follow the relationships between the individuals included in the historical samples, but presumably this can be accomplished using studbook data, which might illuminate relationships between the historical individuals and between historical individuals and the founders of the White Oryx Project in Oman. If the authors could do this, their analyses of kinship could potentially be improved.

There is little overall structure in the current WWR population, though evidence of its historical origins is evident. The management of breeding is an interesting challenge in this species. The recommendation to prioritize pairings between individuals of low kinship (low IBD) may not necessarily optimize retention of overall genetic diversity. Inbred individuals can have unique variation, as is suggested in the analysis of the UAE population. Rather, it would be important to manage allelic loss so as to retain as much overall variation as possible. Altogether, the issue of potential impacts on management can be expanded and clarified.

One concern with SNP calling in Rad-Seq is aligning paralogs, which increases heterozygosity and can give rise to inaccurate analyses of relationships and population diversity. The methods could benefit from greater descriptive detail. While perhaps not generally known, the genome of *O. leucoryx* has a large repetitive component that has been encountered when producing and assembly using short-read data. So, the concern about paralogs is appropriate.

In Table 1 can the studbook numbers of the individuals included in the list of historical samples be provided?

In Table 2, it would be helpful to include a column listing N_e and one for allelic richness.

L 285-6. (“ the main source of variation is mostly within individuals rather than between groups.”) I think the authors may mean to say “ the main source of variation is within groups, rather than between groups.”

One point that would provide perspective in the discussion and be useful for those considering further work is that the Rad-Seq data constitute an explicit data set and any further or expanded analysis would need to repeat work presented in this manuscript. This is in contrast to generating whole genome sequence data which can more readily be compared across experiments, investigators, and time. It would be very important to retain samples, especially perhaps, historical samples for genome sequencing analysis.

The power of the Rad-Seq approach is impacted by sample size which reflects founder effects and bias of ascertainment. In Table 2, it is stated that two individuals from the “USA founders” of historical population were included in the analyses of genetic variation, yet in Table 1, it appears that no historical samples were used for generating the Rad-Seq. Could the authors please explain this apparent discrepancy. Furthermore, since there will be impacts of diversity among the USA founders population depending on their founder contributions and pedigree relationships, can confidence estimates be established for the values in Table 2? The authors might use simulation methods to evaluate the variance in results that come from the Rad-Seq data and generate metrics of confidence and/or significance for the data presented in this study. Without some assessment of the confidence intervals or significance of their findings, the relevance to ongoing management is lessened.

In the mitochondrial DNA haplotype analysis, in the text (l 227-8), "Haplotype E had the lowest frequency (<1%) and was present in a single individual of the WWR-Mix group," but as Figure 3 depicts was present in KSA, USA, and UAE populations. Here is an example of how the manuscript's focus is on the animals in Oman. The perspective might be enlarged, as this is the first and most thorough study of its kind, to reflect the role of the population in the Sultanate of Oman in the global efforts to preserve the genetic variation of this endangered species.

I appreciate the work that went into this study and hope that the findings can be clarified and published.

Decision letter (RSOS-210558.R0)

Dear Ms Brittain

The Editors assigned to your paper RSOS-210558 "Rescued back from extinction in the wild: past, present and future of the genetics of the Arabian oryx in Oman" have now received comments from reviewers and would like you to revise the paper in accordance with the reviewer comments and any comments from the Editors. Please note this decision does not guarantee eventual acceptance.

The reviewers are very positive about your findings and the importance of the work. However, they raise a number of substantive issues, from presentation to analysis, that will require careful consideration. We invite you to respond to the comments supplied below and revise your manuscript. Below the referees' and Editors' comments (where applicable) we provide additional requirements. Final acceptance of your manuscript is dependent on these requirements being met. We provide guidance below to help you prepare your revision.

Please submit your revised manuscript and required files (see below) no later than 21 days from today's (ie 26-Jul-2021) date. Note: the ScholarOne system will 'lock' if submission of the revision is attempted 21 or more days after the deadline. If you do not think you will be able to meet this deadline please contact the editorial office immediately. On this particular occasion, the Editor received several thorough reports, so would be glad to extend the resubmission deadline to allow you to address all the reviewers' comments.

on behalf of Professor Steve Brown (Subject Editor)
 openscience@royalsociety.org

Reviewer comments to Author:

Reviewer: 1

Comments to the Author(s)

The paper represents a significant body of research into the population genetic diversity of Arabian oryx in Oman and implications for future conservation breeding programmes. This species, and particularly its conservation history in Oman, are internationally renowned in conservation reintroductions and this topic will certainly be of broad interest to the conservation community.

The study has been well-designed and the analyses are for the most part appropriate, but the description of the methods causes some confusion over what is being analysed, particularly in terms of the sample groupings.

The description of the results was at times cursory and elsewhere inaccurate. If population genetic analysis is going to be employed and included in the manuscript, the authors need to ensure that it is there for a reason, is accurately presented and adequately explained. Examples of issue are provided in Minor corrections below.

I identified six grammatical corrections / improvements in the abstract. On this basis I suggest that the entire manuscript would benefit from careful revision of the English throughout, by a native speaker. There appear to be native English speakers in the author list who presumably could do this.

While the manuscript needs quite a bit of work to make it suitable for publication, I feel that there is sufficient basis for a publication in RSOS and if the authors are prepared to make the recommended improvements I would be happy to recommend publication.

Minor corrections:

As the references in this journal are simply inserted as numbers, the authors should try to avoid citing just the number when referring to an actual person. For example ‘... approach developed by (49)...’. In this case, please write the name of the author (with et al. if necessary) and place the reference number in parantheses afterwards, e.g. ‘... approach developed by Endelman et al. (49)...’

Line 8 Ethics If laboratory analysis was performed outside of Oman please include details of CITES permits or licences

Abstract:

Line 27 ‘...founders, and private collections...’

Line 29-31 Worth mentioning how oryx came to be in the UAE

Line 35 '...contain...'. Although probably better to use '...display 58% of observed haplotype...'

Line 39 Rephrase. The SNP data did not create anything.

Line 40 '...identification of those individuals for which mating pairs...'

Line 42 Remove 'the'

Line 43 Remove 'the'

Introduction:

Line 49 Rephrase 'a definitive extinction'. Perhaps 'complete extinction'?

Lines 50 & 51 Implies Phoenix zoo is in (on) the Arabian peninsula

Line 66-70 What do the authors mean by 'low' and 'high' levels of genetic diversity? Perhaps insert 'relatively' here, to communicate that different populations are being compared (assuming they are comparable).

Line 93 If known, has there been any mixing of these three herds since they were created?

Line 110 Error in references

Materials and Methods:

Line 113 This implies that some individuals from the original populations (WWR-Oman and WWR-UAE) were not mixed. However at Line 90 the authors suggest that all animals were potentially mixed. Please clarify.

Throughout the results the authors refer to five groups of oryx: Oman founder, USA founder, WWR Mix, WWR Oman and WWR UAE. However it is not clear from Table 1 where the USA founder animals are, as none appear to have been analysed using RAD-seq. This needs clarifying. Please make Table 1 consistent with other Figures and Tables and explain clearly at the start of the Methods, where it states that 3 groups were used (Line 113).

Line 158 What is meant by a 'Bonferroni correction of <math><0.05</math>? What was the P-value employed after Bonferroni correction? How many populations were assumed etc.

Line 179 '...rely on...'

Line 198 It is not clear how an estimate of N_e would inform how different groups have been affected by drift / inbreeding, unless you assume the three (or five?) groups had identical starting conditions. Please explain.

Results:

Line 235 Please clarify whether this is considered to be 58% of extant diversity, or 58% of total diversity ever observed?

Line 241 Reference error

Line 268 onwards Description of PCA results appears confused. PC2 differentiates WWR-UAE, not WWR-Oman as far as I can tell.

Line 284 Fst values. Some of these are actually quite high (>12%) and they vary a lot: <1% to >12%. Have they been tested for significance? More accurate description pairwise FST data required for Table 3

Line 288 Replace 'parentage' with 'familial'

Line 294 Commentary on the usefulness of a group for conservation should be moved to the Discussion. Also, consider that the UAE, although showing greater inbreeding, may also carry distinct genetic diversity.

Discussion:

General: The finding of a distinct mtDNA group is one of the most interesting findings of the work and is probably worthy of more discussion. Do the authors recommend increasing the frequency of these haplotypes in the Oman population. Are there any historical (phylogeographic) explanations for this division? Does it mirror any other patterns of differentiation in Arabian fauna?

Line 313 Would replace 'confirms' with 'indicates'.

Line 325 How do the three groups in the nuclear data compare to the three management herds in Oman? Could the results be due to drift based on isolation of these three herds? If not, is the observed pattern of diversity expected to persist under current management? This is important and more Discussion is required here

Line 377 Sentence doesn't make sense

Figures and Tables:

Figure 1

The Legend is extremely long. Recommend reducing if possible, although the information here is valuable - maybe include long version in Supp. Mat..

The Legend states that the middle boxes show conservation status, but I cannot see this.

Figure 2

Suggest straightening and thinning the black line. Its only needs to join the edges of the relevant areas, not the middle.

The maps apart o have been copied from elsewhere. Include information on the source of the original maps.

The right hand map contains place names that are too small to read. Suggest either removing or making bigger.

Figure 3

Legend box and right hand cluster box are missing sides

Again, the legend text is too long, suggest reducing and placing extra text in Supp. Mat.

Figure 4

Final sentence of the legend refers to identification numbers of individuals on the x-axis, but these are impossible to read and are not critically important - suggest removing.

Table 1 Please align the sample categories with the five 'groups' used throughout the results.

Supp Table 3 Column headers (Sigma, Percentage and Phi) need more explanation. What does this table show?

Supp Table 4 Explain Fh and Falt in the legend

Supp Table 7 How do you define 'highest'?

Reviewer: 2

Comments to the Author(s)

General comments

Using mitochondrial sequences and SNP genotypes generated from RAD-sequencing, the authors analyzed the genetic diversity, inbreeding, structure, and kinship among Arabian oryx maintained at the Al-Wusta Wildlife Reserve in Oman, where an important captive breeding and reintroduction program was established for this species, and among a set of historical samples collected in the 1980s, when the first captive breeding programs for the species were initiated. The animals at the Al-Wusta Wildlife Reserve originate from different sources. Therefore, understanding the genetic contributions and consequences of these sources are useful for the conservation breeding and reintroduction programs, which have lacked formal genetic management strategies until now. The data and results the authors have generated represent an important advancement in our understanding of the genetics of Arabian oryx captive populations and also represents a nice model of how genomic data can be directly implemented to inform breeding recommendations (in the immediate future) and reintroductions (a more distant goal).

The manuscript is generally well-written and presents entirely novel data and results. The amount of text devoted to the introduction, materials and methods, results, and discussion sections is well balanced. The cited literature is comprehensive and up to date. The mitochondrial and RAD-seq data have been analyzed in a comprehensive and rigorous manner, and the authors have appeared to pay attention to sample size issues and the underlying assumptions of the analytical methods they applied. I was especially intrigued with the use of the q-profile approach developed for evolutionary and population genetic applications by Sherwin et al. 2017. This approach standardizes estimates of population genetic parameters into effective measures that make them easily comparable across different sampled groups. The figures and tables are generally of good quality and easily interpretable. The additional results presented in the supplementary figures and tables provide important details that support the conclusions. I especially like Supplementary Figure 6 (the larger heatmap of IBD among all sampled individuals). The manuscript suffers from quite a number of grammatical errors that diminish clarity and readability. I provide comments and recommendations regarding these and other issues for the authors to address, which I think would help improve this important manuscript.

Specific comments

1. Lines 47-48: "IUCN red list of threatened species" This is a formal name and therefore the full name should be capitalized: "IUCN Red List of Threatened Species"
2. Lines 58-59: Sentence awkwardly worded: "The current number of the Arabian oryx is estimated ~1,100 in the wild and ~6,000-7,000 in captivity, zoos, reserves and private collections (1)."

Suggested revision: "The current number of the Arabian oryx is estimated to be ~1,100 in the wild and ~6,000-7,000 in captivity, which includes zoos, reserves and private collections (1)."

3. Lines 78-79: "Eventually, through successful management, the Jalooni population increased to about 400 captive and reintroduced individuals into the wild by 1996 (16)." As written, it's not clear whether the authors' mean a total of 400 individuals from both captive and wild populations were present in 1996 or that by 1996, 400 individuals had been reintroduced into the wild. I assume the former was the intent of the sentence. In that case, remove the phrase "into the wild" to clarify the meaning of the sentence.

4. Line 97: "World Herd" is misspelled in this line.

5. Line 113: "...its primary source population..." As this phrase refers to the "samples" (plural), this should be changed to "...their primary source population..."

6. Line 114: Add a 'the' before "result" in this sentence.

7. Line 122: For the sake of readers, specify the length of the mtDNA CR sequence that was amplified. This is indicated as 638 base pairs in the Results section. However, it's useful to mention this here in the Methods section.

8. Lines 122 and 134: "MtDNA CR" and "mtDNA CR" To improve clarity, these should be qualified by the word 'sequences': "MtDNA CR sequences" and "mtDNA CR sequences"

9. Line 131: Specify the method and instrument used to quantify libraries. I assume using a qPCR?

10. Line 152: "...to obtain equal length of sequences..." Delete the "of" in this sentence.

11. Lines 154-155: What to the authors mean by "individuals sampled more than once"? Were some individuals sequenced more than once in the RAD-seq protocol? If so, why and what proportion of the 135 samples used for the RAD-seq analyses?

12. Line 172: "explain" should be changed to "explains" in this sentence.

13. Line 179: "PCA is a technique that does not relay in any genetic assumption..." I assume the authors meant "'PCA is a technique that does not rely on any genetic assumption...'"?

14. Line 187: "...to be used in samples of unequal sample sizes..." To improve clarity, change to "...to be used among groups of unequal sample sizes..."

15. Lines 189-190: "...to allow a direct comparison." A comparison of what, exactly? Among groups? Populations? Please specify.

16. Line 191: Add a "the" before "Shannon index."

17. Line 192: Add a "a" before "measure."

18. Lines 194-195: "To identify the group contributing with most to the variance of genetic variation..." Suggested revision: "To identify the groupings contributing the most to the variance of genetic variation..."

19. Line 216: Delete "Yang et al." in this sentence.

20. Line 221: "MtDNA control region (638 bp) was generated..." To improve clarity and grammar, change to: "MtDNA control region sequences 638 bp in length were generated..."
21. Line 225: "Haplotypes sequences are available in GenBank..." Suggested revision: "Haplotype sequences are available from GenBank..."
22. Lines 238-239: In Figure 3: these two haplogroups are referred to as Cluster I and Cluster II. For the sake of consistency, the authors should select one of these - haplogroup or cluster.
23. Line 239: I think the authors mean haplogroup (or Cluster) I here.
24. Line 252: Change "no present" to "not present" in this sentence.
25. Line 254: Add "was" before "comprised" in this sentence.
26. Line 255: Add a "the" before "National Center for Biotechnology Information"
27. Line 259: Awkward wording: "The inferred number of populations (K) that explained the best the genetic structure..." Suggested revision: "The inferred number of populations (K) that best explained the genetic structure..."
28. Line 262: Add a comma after "likelihood" in this sentence.
29. Line 272: "founders/descendent." To improve clarity, this should be changed to "founders and descendents"
30. Line 298: Change "less inbred" to "least inbred"
31. Line 300: Remove the comma after "PCA" in this sentence.
32. Lines 314-315: Awkward sentence construction: "The additional four haplotypes were identified in the WWR individuals sourced from the Arabian Peninsula including KSA and UAE..." Suggested revision: "The additional four haplotypes identified in the WWR individuals were sourced from the Arabian Peninsula, including KSA and UAE..."
33. Line 333: Change "recently" to "recent"
34. Line 343: Add a comma after "group" in this sentence.
35. Lines 351-352: Awkward sentence construction: "After, acknowledging the above caveats, overall our results showed..." Suggested revision: "After acknowledging the above caveats, our results showed..."
36. Line 366: it's not clear what the authors mean about $q=0$ having "sampling problems." Do they mean that the estimation $q=0$ is influenced by sample size of populations measured? Please revise to specify the problems associated with the q-profile approach.
37. Add a comma after "Together" in this sentence.
38. Line 382: Change "by" to "in" - "in the USA founder group."
39. Line 386: Change "use" to "used" in this sentence.

40. Lines 396-397: "...aimed to maximize genetic diversity and control the negative effects of inbreeding and evolutionary potential." How is controlling evolutionary potential a negative effect? I think the authors meant to write: "...aimed to maximize genetic diversity and evolutionary potential and control the negative effects of inbreeding."

41. Line 413: Add a comma after "Arabian Peninsula" in this sentence.

42. Line 415: "long" should be changed to "long-term" in this sentence.

43. Line 417: "separating offspring" - the authors need to indicate when such a separation would occur. I assume after weaning? Or when individuals reach sexual maturity. Such a recommendation has important implications and therefore needs to be better specified.

44. Lines 421-424: Awkward sentence construction: "...establishing satellite breeding groups, this could be smaller breeding nucleus with multiple females complemented with a seasonal rotational male breeding scheme so females could had the opportunity to breed with other representatives males (63) from the genetic lineages identified in this study;" Suggested revision: "...establishing satellite breeding groups, which would include a smaller breeding nucleus with multiple females, complemented with a seasonal rotational male breeding scheme so that females have the opportunity to breed with other representative males (63) from the genetic lineages identified in this study;"

45. Lines 429-430: "from a continuing genetic monitoring program" Do the authors mean using the same genetic approaches as reported in their paper? Please specify, as this will provide useful guidance to all managers of Arabian oryx breeding and reintroduction programs.

46. Line 436: Change "samples" to "sample" in this sentence.

47. Figure 1: Add the word "program" after "reintroduction" in the first sentence of the caption. For the middle boxes of the figure - first middle box: "Conducted of "Operation Oryx" should be revised to "Operation Oryx" conducted". In the fourth middle box, text can be changed to "Reintroduction of "Oman-Herd into the wild."

In the bottom boxes, "Operation Oryx" in the third box should be capitalized for the sake of consistency. Similarly, "World-herd" can be changed to "World Herd." in the fourth box.

48. Figure 2: Awkward wording: "Jalooni located in west north of Al-Wusta Wildlife Reserve." Suggested revision: "Jalooni is located in northwestern part of the Al-Wusta Wildlife Reserve." Also, please specify what the purple blotches indicate in the map figure on the left. I assume these are the Arabian oryx reserves in Israel, Kuwait, KSA, Oman and UAE?

49. Figure 3: Change the period to a comma before "Jordan and USA (KU985184, Ochoa et al., 2016)" in the caption. Also, add a comma before "with seven haplotypes (A-G)..." In addition, revise the following sentence: "Further details on the distribution of these haplotypes among other the captive and reintroduction programs in the Arabian Peninsula..." as "Further details on the distribution of these haplotypes among the other captive and reintroduction programs in the Arabian Peninsula..."

50. Figure 4: The clustering plot and individual ID numbers at the bottom need to be increased in size to improve readability. In the current version, the IDs are too small to be read.

51. Figure 7 caption: "Endelman & Jannink19" should be changed to "Endelman & Jannink (2012)"

52. Table 1: Are specific years in the 1980s available for the historical samples?

53. Tables 2 and 3: Can the authors test for significant difference in the values (e.g., Shannon index, F_{st}) among the groups reported in these tables?

Reviewer: 3

Comments to the Author(s)

See attached review file ("Royal Society Review 210558.pdf").

Reviewer: 4

Comments to the Author(s)

RSOS-210558

This manuscript presents a very interesting and worthwhile contribution to the literature on a species that was critically endangered, thought to be extinct in the wild, had its population expanded through managed breeding under human care, was released back into the wild and remains a focal species for private collections. The assessments of genetic diversity previously performed were typically for subpopulations of the overall population, lacked resolving power and made only minor contributions to population management regarding retention of the greatest possible extent of the surviving genetic variation of the species, the Arabian oryx, or white oryx, *Oryx leucoryx*. As the authors state (l 88-90), "The breeding program focused on increasing population size, rather than on increasing genetic diversity." and (l 99-101), "Additionally, the absence of genetic data has limited the ability of this program to move from a strategy of random mating to a genetic-based (or group-based) management."

To address the information missing from what has become standard population management practice for conservation of large mammal species, the authors obtained a variety of samples of blood, tissues, or DNA extracts from previous studies and conducted two separate studies. One investigated mitochondrial DNA diversity using PCR amplification and DNA sequencing of 174 individuals, including samples from individuals from the early generations of the breeding efforts. The other used reduced representation library sequencing methods (Rad-Seq) to interrogate presumptively homologous portions of the *O. leucoryx* genome from 135 individuals in the later generations of the breeding efforts.

Usable results were obtained for the mitochondrial analyses from more samples than for the Rad-Seq analysis, a not unexpected result as the authors explained (l 119), "we kept the samples with high DNA quality only."

A mitochondrial DNA network was generated and, using comparison between the 'historical' samples (from the early generations of captive breeding when a pedigree for the World Herd was being kept), and the samples from the recent generations of animals bred in the Sultanate of Oman the authors show that this population (WWR) as a whole contains 58% of the global mtDNA 236 haplotype diversity. There are some indications of structure in the mitochondrial DNA haplotype map, but the total branch length of the network is relatively small and the authors identify two clusters of haplotypes that are separated by seven mutations.

The results of the Rad-Seq analysis are presented several ways: as a PCA plot, a Bayesian cluster analysis and with a heat-map of estimated relatedness. These data and analyses of F_{st} combine to suggest that there was modest genetic differentiation in the groups of individuals that have been incorporated into the current managed population in the Sultanate of Oman. Although admixture

has taken place between the combined groups, there is still some structure within the overall Oman population.

Management goals are alluded to, but the paper might have more impact if concrete examples were given.

Disparate founder contributions can have important impacts in analyses like this one when attempting to manage allele loss. It is hard to follow the relationships between the individuals included in the historical samples, but presumably this can be accomplished using studbook data, which might illuminate relationships between the historical individuals and between historical individuals and the founders of the White Oryx Project in Oman. If the authors could do this, their analyses of kinship could potentially be improved.

There is little overall structure in the current WWR population, though evidence of its historical origins is evident. The management of breeding is an interesting challenge in this species. The recommendation to prioritize pairings between individuals of low kinship (low IBD) may not necessarily optimize retention of overall genetic diversity. Inbred individuals can have unique variation, as is suggested in the analysis of the UAE population. Rather, it would be important to manage allelic loss so as to retain as much overall variation as possible. Altogether, the issue of potential impacts on management can be expanded and clarified.

One concern with SNP calling in Rad-Seq is aligning paralogs, which increases heterozygosity and can give rise to inaccurate analyses of relationships and population diversity. The methods could benefit from greater descriptive detail. While perhaps not generally known, the genome of *O. leucoryx* has a large repetitive component that has been encountered when producing and assembly using short-read data. So, the concern about paralogs is appropriate.

In Table 1 can the studbook numbers of the individuals included in the list of historical samples be provided?

In Table 2, it would be helpful to include a column listing N_e and one for allelic richness.

L 285-6. (" the main source of variation is mostly within individuals rather than between groups.") I think the authors may mean to say " the main source of variation is within groups, rather than between groups."

One point that would provide perspective in the discussion and be useful for those considering further work is that the Rad-Seq data constitute an explicit data set and any further or expanded analysis would need to repeat work presented in this manuscript. This is in contrast to generating whole genome sequence data which can more readily be compared across experiments, investigators, and time. It would be very important to retain samples, especially perhaps, historical samples for genome sequencing analysis.

The power of the Rad-Seq approach is impacted by sample size which reflects founder effects and bias of ascertainment. In Table 2, it is stated that two individuals from the "USA founders" of historical population were included in the analyses of genetic variation, yet in Table 1, it appears that no historical samples were used for generating the Rad-Seq. Could the authors please explain this apparent discrepancy. Furthermore, since there will be impacts of diversity among the USA founders population depending on their founder contributions and pedigree relationships, can confidence estimates be established for the values in Table 2? The authors might use simulation methods to evaluate the variance in results that come from the Rad-Seq data and generate metrics of confidence and/or significance for the data presented in this study. Without some assessment of the confidence intervals or significance of their findings, the relevance to ongoing management is lessened.

In the mitochondrial DNA haplotype analysis, in the text (1 227-8), "Haplotype E had the lowest frequency (<1%) and was present in a single individual of the WWR-Mix group," but as Figure 3 depicts was present in KSA, USA, and UAE populations. Here is an example of how the manuscript's focus is on the animals in Oman. The perspective might be enlarged, as this is the first and most thorough study of its kind, to reflect the role of the population in the Sultanate of Oman in the global efforts to preserve the genetic variation of this endangered species.

I appreciate the work that went into this study and hope that the findings can be clarified and published.

===PREPARING YOUR MANUSCRIPT===

===PREPARING YOUR REVISION IN SCHOLARONE===

Author's Response to Decision Letter for (RSOS-210558.R0)

See Appendix B.

RSOS-210558.R1 (Revision)

Review form: Reviewer 2

Is the manuscript scientifically sound in its present form?

Yes

Are the interpretations and conclusions justified by the results?

Yes

Is the language acceptable?

Yes

Do you have any ethical concerns with this paper?

No

Have you any concerns about statistical analyses in this paper?

No

Recommendation?

Accept as is

Comments to the Author(s)

I have read the revised manuscript by Rawahi et al. and find that it has been substantially improved compared to the original submission. The authors have done an excellent job in providing thorough and satisfactory responses to the reviewers' comments, including my own, and have revised the manuscript accordingly. The streamlined text and corrected writing have improved the readability and presentation of the manuscript. I recommend this first-rate manuscript be accepted for publication.

Review form: Reviewer 3

Is the manuscript scientifically sound in its present form?

Yes

Are the interpretations and conclusions justified by the results?

Yes

Is the language acceptable?

Yes

Do you have any ethical concerns with this paper?

No

Have you any concerns about statistical analyses in this paper?

No

Recommendation?

Accept with minor revision (please list in comments)

Comments to the Author(s)

Review – Royal Society RSOS-210558.R1

Overall comments to authors:

- The aim of this paper is to use population genetic tools, using data obtained from one mitochondrial gene and genome-wide SNP genotyping completed using ddRAD-sequencing, to describe the levels of genetic diversity and genetic management history of the Arabian oryx in Oman. The authors use a number of analytical techniques to calculate multiple population genetic measures and provide implications for future genetic management based on the results of past management.
- The authors have done an excellent job responding to the suggestions and edits provided by the four reviewers in the first review phase. Parts of the analyses are more clear and it is evident that much work has gone into adding to the discussion to make the results relevant and applicable to future management. The main suggestion is that much of the new content in the Introduction, Discussion, Tables, and Supplementary information needs more editing for clarity, ease of reading, and for more detail. A number of suggestions have been provided below in the section-by-section comments.

Section by Section comments:

Abstract

- Line 21: It is unclear what “WWR” program is at this point in abstract.
- Line 24: Need space before sentence starting “We...”
- Line 26: “three different groups” could have more explanation. Groups of what?
- Line 28: should read “contribute to maximizing genetic diversity.”

Introduction

- Line 46-47: Awkward phrasing of sentence “captive breeding was not aimed to minimize...”. Rephrase to say something like “captive breeding was not implemented using individual genetic management through minimizing population mean kinship”.
- Lines 52-56: The reviewer appreciates the addition of supplementary Table 1 and the detail put into this. However, at this point in the introduction it is still unclear what “low levels of genetic diversity” and “high inbreeding and relatedness” mean to the authors. While a number of papers have been added and cited and the reader is suggested to review the Supplementary Table, it would be very helpful to the reader to be given some of the numbers to understand the context of what is meant by low and high in previous literature here in the introduction.
- Line 56: Rephrase start of this sentence to read something like “In addition to the research completed using microsatellites, 12 mtDNA control region haplotypes....”.
- Line 57: include reference to Supplementary Table 1 in addition to Supplementary Table 2.
- Line 85: add reference/citation about random mating and maintenance of genetic diversity
- Line 87-88: Difficult to understand phrasing “has acknowledge the importance...”, unsure of the aim of this sentence.

Materials and Methods

- Line 95-96: Please check reference/citation, as there are errors in the citation software and how these are showing to the reader.

Results

- Line 221: this should be broken up into two sentences. “This haplotype...” is the start of a new sentence.

- Line 231 and 260: Please check reference/citation, as there are errors in the citation software and how these are showing to the reader.
- Line 292-293: It is unclear what “above analyses” were altered with the changed assignments after re-assigning the three incorrectly identified individuals. In addition, it might make more sense in the flow of results to put this result earlier in this section so that you can confirm that all downstream analyses (whatever they may be!) were done with these corrected identifications.

Discussion

- Line 339: wording is awkward, should read “current sequencing technologies allow for genotyping thousands of markers...”
- Line 349-351: The authors present a summary of their SNP results and demonstration of an increase in heterozygosity in the WWR-Mix group and say that this is an interesting result. However, there is not interpretation as to why this interesting or not expected. Please provide more discussion on this point.
- Line 361-362: wording “history of frequent founder effects” – this phrasing is a bit confusing. Please clarify what you mean here related to admixture and movement of individuals related to the q-profile and FST results.
- Line 364-365: Awkward sentence structure or not a sentence.
- Line 365-367: These sentences also need to be edited. End sentence after “our results demonstrate the advantage of using the q-profile”, then provide examples in the follow up sentences.
- Line 381: missing word? “such as the data generated in this study?”
- Line 382-384: should read “clear strategy aimed at maximizing genetic diversity and evolutionary potential and minimizing the effects of inbreeding”.

Other

- Figure 6: This was difficult to read as there were black bars in the pdf version over the x and y axes
- Figure 7: Visually, the x-axis is off to the left a little bit making the ID numbers not quite line up in the heat map. Is the black bar in the figure meant to be there? Might be good to remove that if possible.
- Table 2: Would be nice to format this table so that headings are not split across multiple rows as well as so the values in each row. This table is very difficult to read as it is currently formatted. In addition, the Table description/header is missing.
- Table 3: The last column says that all p-values are greater than 0.01. Did you mean less than? If so the sign needs to be reversed, and if you do mean greater than, you should provide the value. In addition, the Table description/header is missing.
- Supplementary Figure 1: the “k” in the table of haplotypes is lowercase and all other letters are uppercase.
- Supplementary Figure 2: the figure legend has the X- and Y-axis label information reversed in the description.
- Supplementary Figure 6: would be nice to add the same figure legend and more detailed description here that is in the paper Figure 7, particularly since the individual IDs will be difficult or impossible to read in this figure, and might be blurry when zooming in, the reader will still want to get as much information out of the figure as possible.
- Supplementary Figure 7: The caption for this figure could provide more details about how these individuals were originally determined to be assigned to the incorrect group. What was the process that was used to identify these individuals?
- Supplementary Figure 8: The caption for this figure could provide more details so that I can stand alone. What is meant here by three management herds (enclosures)? The reader needs to go find the details in the main paper to understand what the three herds are in relationship to the WWR-Mix herd so more detail is needed.

Review form: Reviewer 4

Is the manuscript scientifically sound in its present form?

Yes

Are the interpretations and conclusions justified by the results?

Yes

Is the language acceptable?

Yes

Do you have any ethical concerns with this paper?

No

Have you any concerns about statistical analyses in this paper?

No

Recommendation?

Accept as is

Comments to the Author(s)

The revision responds adequately to the comments of reviewers.

Decision letter (RSOS-210558.R1)

Dear Ms Brittain

On behalf of the Editors, we are pleased to inform you that your Manuscript RSOS-210558.R1 "Rescued back from extinction in the wild: past, present and future of the genetics of the Arabian oryx in Oman" has been accepted for publication in Royal Society Open Science subject to minor revision in accordance with the referees' reports. Please find the referees' comments along with any feedback from the Editors below my signature.

All of the reviewers are very positive about the improvements to the manuscript. One reviewer recommends a number of minor improvements. We invite you to respond to the comments and revise your manuscript. Below the referees' and Editors' comments (where applicable) we provide additional requirements. Final acceptance of your manuscript is dependent on these requirements being met. We provide guidance below to help you prepare your revision.

Please submit your revised manuscript and required files (see below) no later than 7 days from today's (ie 24-Jan-2022) date. Note: the ScholarOne system will 'lock' if submission of the revision is attempted 7 or more days after the deadline. If you do not think you will be able to meet this deadline please contact the editorial office immediately.

on behalf of Steve Brown (Subject Editor)
openscience@royalsociety.org

Associate Editor Comments to Author (Professor Steve Brown):

Comments to the Author:

The manuscript is much approved and can be accepted after dealing with the minor suggestions of the reviewer

Reviewer comments to Author:

Reviewer: 3

Comments to the Author(s)

Review – Royal Society RSOS-210558.R1

Overall comments to authors:

- The aim of this paper is to use population genetic tools, using data obtained from one mitochondrial gene and genome-wide SNP genotyping completed using ddRAD-sequencing, to describe the levels of genetic diversity and genetic management history of the Arabian oryx in Oman. The authors use a number of analytical techniques to calculate multiple population genetic measures and provide implications for future genetic management based on the results of past management.
- The authors have done an excellent job responding to the suggestions and edits provided by the four reviewers in the first review phase. Parts of the analyses are more clear and it is evident that much work has gone into adding to the discussion to make the results relevant and applicable to future management. The main suggestion is that much of the new content in the Introduction, Discussion, Tables, and Supplementary information needs more editing for clarity, ease of reading, and for more detail. A number of suggestions have been provided below in the section-by-section comments.

Section by Section comments:

Abstract

- Line 21: It is unclear what “WWR” program is at this point in abstract.
- Line 24: Need space before sentence starting “We...”
- Line 26: “three different groups” could have more explanation. Groups of what?
- Line 28: should read “contribute to maximizing genetic diversity.”

Introduction

- Line 46-47: Awkward phrasing of sentence “captive breeding was not aimed to minimize...”. Rephrase to say something like “captive breeding was not implemented using individual genetic management through minimizing population mean kinship”.
- Lines 52-56: The reviewer appreciates the addition of supplementary Table 1 and the detail put into this. However, at this point in the introduction it is still unclear what “low levels of genetic diversity” and “high inbreeding and relatedness” mean to the authors. While a number of papers have been added and cited and the reader is suggested to review the Supplemental Table, it would be very helpful to the reader to be given some of the numbers to understand the context of what is meant by low and high in previous literature here in the introduction.
- Line 56: Rephrase start of this sentence to read something like “In addition to the research completed using microsatellites, 12 mtDNA control region haplotypes....”.
- Line 57: include reference to Supplementary Table 1 in addition to Supplementary Table 2.
- Line 85: add reference/citation about random mating and maintenance of genetic diversity
- Line 87-88: Difficult to understand phrasing “has acknowledge the importance...”, unsure of the aim of this sentence.

Materials and Methods

- Line 95-96: Please check reference/citation, as there are errors in the citation software and how these are showing to the reader.

Results

- Line 221: this should be broken up into two sentences. “This haplotype...” is the start of a new sentence.
- Line 231 and 260: Please check reference/citation, as there are errors in the citation software and how these are showing to the reader.
- Line 292-293: It is unclear what “above analyses” were altered with the changed assignments after re-assigning the three incorrectly identified individuals. In addition, it might make more sense in the flow of results to put this result earlier in this section so that you can confirm that all downstream analyses (whatever they may be!) were done with these corrected identifications.

Discussion

- Line 339: wording is awkward, should read “current sequencing technologies allow for genotyping thousands of markers...”
- Line 349-351: The authors present a summary of their SNP results and demonstration of an increase in heterozygosity in the WWR-Mix group and say that this is an interesting result. However, there is not interpretation as to why this interesting or not expected. Please provide more discussion on this point.
- Line 361-362: wording “history of frequent founder effects” – this phrasing is a bit confusing. Please clarify what you mean here related to admixture and movement of individuals related to the q-profile and FST results.
- Line 364-365: Awkward sentence structure or not a sentence.
- Line 365-367: These sentences also need to be edited. End sentence after “our results demonstrate the advantage of using the q-profile”, then provide examples in the follow up sentences.
- Line 381: missing word? “such as the data generated in this study?”
- Line 382-384: should read “clear strategy aimed at maximizing genetic diversity and evolutionary potential and minimizing the effects of inbreeding”.

Other

- Figure 6: This was difficult to read as there were black bars in the pdf version over the x and y axes

- Figure 7: Visually, the x-axis is off to the left a little bit making the ID numbers not quite line up in the heat map. Is the black bar in the figure meant to be there? Might be good to remove that if possible.
- Table 2: Would be nice to format this table so that headings are not split across multiple rows as well as so the values in each row. This table is very difficult to read as it is currently formatted. In addition, the Table description/header is missing.
- Table 3: The last column says that all p-values are greater than 0.01. Did you mean less than? If so the sign needs to be reversed, and if you do mean greater than, you should provide the value. In addition, the Table description/header is missing.
- Supplementary Figure 1: the “k” in the table of haplotypes is lowercase and all other letters are uppercase.
- Supplementary Figure 2: the figure legend has the X- and Y-axis label information reversed in the description.
- Supplementary Figure 6: would be nice to add the same figure legend and more detailed description here that is in the paper Figure 7, particularly since the individual IDs will be difficult or impossible to read in this figure, and might be blurry when zooming in, the reader will still want to get as much information out of the figure as possible.
- Supplementary Figure 7: The caption for this figure could provide more details about how these individuals were originally determined to be assigned to the incorrect group. What was the process that was used to identify these individuals?
- Supplementary Figure 8: The caption for this figure could provide more details so that I can stand alone. What is meant here by three management herds (enclosures)? The reader needs to go find the details in the main paper to understand what the three herds are in relationship to the WWR-Mix herd so more detail is needed.

Reviewer: 2

Comments to the Author(s)

I have read the revised manuscript by Rawahi et al. and find that it has been substantially improved compared to the original submission. The authors have done an excellent job in providing thorough and satisfactory responses to the reviewers' comments, including my own, and have revised the manuscript accordingly. The streamlined text and corrected writing have improved the readability and presentation of the manuscript. I recommend this first-rate manuscript be accepted for publication.

Reviewer: 4

Comments to the Author(s)

The revision responds adequately to the comments of reviewers.

===PREPARING YOUR MANUSCRIPT===

one version should clearly identify all the changes that have been made (for instance, in coloured highlight, in bold text, or tracked changes);

===PREPARING YOUR REVISION IN SCHOLARONE===

- If you are providing image files for potential cover images, please upload these at this step, and inform the editorial office you have done so. You must hold the copyright to any image provided.
- A copy of your point-by-point response to referees and Editors. This will expedite the preparation of your proof.

- Ensure that your data access statement meets the requirements at <https://royalsociety.org/journals/authors/author-guidelines/#data>. You should ensure that you cite the dataset in your reference list. If you have deposited data etc in the Dryad repository, please only include the 'For publication' link at this stage. You should remove the 'For review' link.
- If you are requesting an article processing charge waiver, you must select the relevant waiver option (if requesting a discretionary waiver, the form should have been uploaded, see 'File upload' above).
- If you have uploaded any electronic supplementary (ESM) files, please ensure you follow the guidance at <https://royalsociety.org/journals/authors/author-guidelines/#supplementary-material> to include a suitable title and informative caption. An example of appropriate titling and captioning may be found at https://figshare.com/articles/Table_S2_from_Is_there_a_trade-off_between_peak_performance_and_performance_breadth_across_temperatures_for_aerobic_scope_in_teleost_fishes_/3843624.

Author's Response to Decision Letter for (RSOS-210558.R1)

See Appendix C.

Decision letter (RSOS-210558.R2)

Dear Ms Brittain,

I am pleased to inform you that your manuscript entitled "Rescued back from extinction in the wild: past, present and future of the genetics of the Arabian oryx in Oman" is now accepted for publication in Royal Society Open Science.

on behalf of Professor Steve Brown (Associate Editor) and Steve Brown (Subject Editor)
openscience@royalsociety.org

Appendix A

Review – Royal Society RSOS-210558

Overall comments to authors:

- The aim of this paper is to use population genetic tools, using data obtained from one mitochondrial gene and genome-wide SNP genotyping, to describe the levels of genetic diversity and genetic management history of the Arabian oryx in Oman. The authors use a number of analytical techniques to calculate multiple population genetic measures and provide implications for future genetic management based on the results of past management.
- This is a well-written paper, with appropriate references and all methods and results are easy to read and interpret. The authors are to be commended on this submission, which will add to the growing body of literature using molecular genetic and empirical data to manage *ex situ* populations. The main suggestion is for the authors to expand the discussion, particularly in the management implications suggestion, to be more specific in their recommendations for genetic management. They should spend some time discussing both the feasibility of their recommendations, but also the differences and trade-offs in future genetic implications when applying the individual genetic management recommendations compared to more group or herd-level recommendations. The authors make suggestions that are more intensive and would require more intensive data collection and pedigree tracking and are more useful in an individual-management approach, verses those that are less intensive such as rotating breeding males, which are somewhat less intensive. This part of the discussion felt more like a list of options, rather than considering their results to make specific recommendations and considering the genetic implications of each recommendation.

Section by Section comments:

Abstract

- Line 25: It would be more clear on the dates of events to explicitly state that the reintroduction commenced a decade after the species was declared extinct in the wild (1982 specifically, or 1980 as in Figure 1).

Introduction

- Line 63: Instead of simply “captive breeding was not managed through studbooks”, expand upon this to state something to the effect of “captive breeding was not implemented using typical individual genetic management through minimizing population mean kinship”. The studbook is the database to maintain data, but is not the tool for management.
- Lines 64-70: It would be helpful to include more specific genetic statistics from these papers to give the reader an idea of what is meant by “low levels of genetic diversity” and “high levels of inbreeding and relatedness” relative to the “high level of genetic diversity” found in the Saudi Arabian group.

- Lines 72-73: Could the authors provide the captive populations that were sampled for these mtDNA studies to allow the reader to understand where the 12 haplotypes were identified?
- Lines 73-75: It would help to expand upon this statement that the data provided in the previous studies could not be used for intensive individual-based management or individual breeding strategies. The information provided at the population or herd level in previous studies could be used to develop some basic group or herd-management style recommendations.
- Line 79: needs clarification. Do you mean the herd grew to 400 individuals and then those individuals were used for reintroduction? Or that on the WWR there were multiple herds delineated by those consisting of captive animals and those containing reintroduced animals? The reviewer believes it is the latter, but more details or explanation in the sentence would help with this and reference to Figure 1 and Table 1 would also clarify this.
- Line 94: instead of “breeding strategy”, suggest change to “genetic management strategy”.
- Lines 100-101: Rephrase, as random mating is one type of management that can work to maintain genetic diversity (and is often a strategy that is better than non-random management without genetic or pedigree data). Likely, the authors intention is to discuss mean kinship based management, which requires a mostly known pedigree and is an individual-based management strategy. Group management can also be done via group or herd kinship, but this specific group management strategy does not appear to be the aim of the authors and the reason for collecting these genetic data.

Materials and Methods

- Line 110-111: Please check reference/citation.
- Lines 112-114: Please provide more details as to how samples were known to be from each lineage. The introduction states that no studbook or records were maintained, but some information was available to do this pre-sorting into an assumed lineage.
- Line 179: Should this say “rely on” instead of “relay in”?

Results

- Line 239: Should this be “within haplotype I”?
- Line 240 and 271: Please check reference/citation.
- Line 252: Correct to “loci NOT present in all groups”
- Line 259: extra “the”

Discussion

- Line 333: “recently” should be “recent”
- Line 347-349: This sentence sets up for the authors to provide a number of reasons that SNP derived data are different from microsatellites, but then the authors provide a single reason. Are there any additional reasons to cite? Line 352 mentions multiple caveats so multiple reasons should be provided.

- Line 387: The authors state that they provide “molecular and relatedness data” in the current study but it would help to make this sentence more specific to the methods and results and that the authors generated molecular data to calculate X, Y, Z population genetics statistics (not simply relatedness data, which are not explicitly provided in the results).
- Line 388-389: It would be helpful for the reader for the authors to expand upon specifically, but briefly, what they mean by “management of this program is informed by genetics with defined goals”. This would strengthen the lead into the subsequent section on management implications.
- Line 396-397: rephrase sentence structure, right now it reads as if you are suggesting “controlling the negative effects of...evolutionary potential” but likely you mean to suggest to use genetic management to maintain evolutionary potential.
- Line 399-401, 418-420 and the entire section on management implications: Reconstruction of pedigree is one option for future management. Given that the parentage assignment results only uncovered 20 first- and second-order relationships to allow for reconstruction, and the relatedness results were only briefly discussed and not used in the results to develop a system of evaluating individuals’ empirical kinships for future pedigree and thus individual management, it may strengthen this paper to use this section to discuss the ability of the managers in these herds to follow through with management recommendations that are more individual based (as recommended in these lines) versus a strategy that uses maximum avoidance of inbreeding, or even more group or herd based methods using the results (Figure 7 as an example) to recommend transfer of individuals into different herds which would improve diversity (as you touched on in the sentence on rotational male breeding). The results in the paper already demonstrate that some of the “random” management strategies have proven successful at maintaining diversity and reducing inbreeding in some herds, so a discussion and separation of the recommendations that are more intensive from a management and data collection perspective, to those that are less intensive would be beneficial. As the *ex situ* management community aims to demonstrate the benefits and successes of less-intense genetic management strategies, this paper has the data and results to contribute to that discussion and body of literature.

Other

- Figure 1: “establishment” spelled incorrectly in caption and some other small typos. Thank you for the inclusion of this timeline and diagram to help orient the reader to the history of the moves and establishment of the different herds and locations.
- Supplementary Table 1: It is a bit unclear the distinction between Historical and Published sources so providing a bit more detail in the table description would help clarify this.

Appendix B

Reviewer: 1

Comments to the Author(s)

The paper represents a significant body of research into the population genetic diversity of Arabian oryx in Oman and implications for future conservation breeding programmes. This species, and particularly its conservation history in Oman, are internationally renowned in conservation reintroductions and this topic will certainly be of broad interest to the conservation community.

*The study has been well-designed and the analyses are for the most part appropriate, but the description of the methods causes some confusion over what is being analysed, particularly in terms of the sample groupings.

Description about the grouping of sample was improved throughout the manuscript.

*The description of the results was at times cursory and elsewhere inaccurate. If population genetic analysis is going to be employed and included in the manuscript, the authors need to ensure that it is there for a reason, is accurately presented and adequately explained. Examples of issue are provided in Minor corrections below.

Description of the results were improved following the reviewer's recommendations.

I identified six grammatical corrections / improvements in the abstract. On this basis I suggest that the entire manuscript would benefit from careful revision of the English throughout, by a native speaker. There appear to be native English speakers in the author list who presumably could do this.

Grammatical issues were corrected and revision has been performed by a native speaker.

While the manuscript needs quite a bit of work to make it suitable for publication, I feel that there is sufficient basis for a publication in RSOS and if the authors are prepared to make the recommended improvements I would be happy to recommend publication.

Minor corrections:

*As the references in this journal are simply inserted as numbers, the authors should try to avoid citing just the number when referring to an actual person. For example '... approach developed by (49)...'. In this case, please write the name of the author (with et al. if necessary) and place the reference number in parantheses afterwards, e.g. '... approach developed by Endelman et al. (49)...'

All the references have been formatted correctly as suggested.

*Line 8 Ethics If laboratory analysis was performed outside of Oman please include details of CITES permits or licences

Permit information is now reported in the Acknowledgement section, as follows:

"Oryx blood and DNA samples were exported to the UK under import permit from the Office for Conservation of the Environment Oman (CITES Certificate no. 172/2015) and under import permit from the UK (CITES Certificate no. 540806/01)."

Abstract:

*Line 27 ‘...founders, and private collections...’

Sentence was modified as suggested:

“The reintroduction of this species into the wild commenced a decade later from two main sources: the “World Herd” established at the Phoenix Zoo from a small number of founders, and private collections in Saudi Arabia”.

*Line 29-31 Worth mentioning how oryx came to be in the UAE

We added the following text in the abstract:

“In UAE only Abu Dhabi and Dubai are the only emirates known to have Arabian oryx, distributed among private collections and the Al Ain Zoo. The collection of this Zoo was founded around 1975 with a male and two females, allegedly wild caught.”

*Line 35 ‘...contain...’. Although probably better to use ‘...display 58% of observed haplotype...’

Sentence was modified as suggested:

“Mitochondrial results showed the individuals from Al-Wusta Wildlife Reserve display 58% of observed haplotype diversity found in this species”.

*Line 39 Rephrase. The SNP data did not create anything.

The sentence was modified to be clearer:

“Pedigree information obtained from SNP data, allowed us to identify those individuals for which mating pairs and groups could most effectively contribute to maximising genetic diversity”.

*Line 40 ‘...identification of those individuals for which mating pairs...’

The sentence was modified as suggested:

Pedigree information obtained from SNP data, allowed us to identify those individuals for which mating pairs and groups could most effectively contribute to maximising genetic diversity.

*Line 42 Remove ‘the’

“the” was removed

*Line 43 Remove ‘the’

“the” was removed

Introduction:

*Line 49 Rephrase 'a definitive extinction'. Perhaps 'complete extinction'?

The sentence was modified as suggested:

“However, conservation efforts rescued the Arabian oryx from a complete extinction...”

*Lines 50 & 51 Implies Phoenix zoo is in (on) the Arabian peninsula.

Sentence was modified to avoid confusion:

“Conservation efforts in other parts of the world included the establishment of the “World Herd” in the Phoenix Zoo (USA)”

*Line 66-70 What do the authors mean by 'low' and 'high' levels of genetic diversity? Perhaps insert 'relatively' here, to communicate that different populations are being compared (assuming they are comparable).

The word relatively has been added in:

“Genetic studies have found relatively low levels of population differentiation...” and

“More recent microsatellite studies of current captive breeding programs in Jordan, the United Arab Emirates (UAE) and Qatar found relatively low levels of genetic diversity...”

*Line 93 If known, has there been any mixing of these three herds since they were created?

Individuals are rotated at random between the three enclosures. We added this information as follows:

“Animals were divided randomly irrespective of their source population and/or parental relationships and have been kept in three enclosures (ranging from 0.27 to 0.74 Km²), however individuals are rotated at random between the three enclosures.”

*Line 110 Error in references

Reference was corrected:

“Blood samples were obtained from 138 randomly selected individuals at WWR (Fig. 2; Animal Welfare Committee at the Animal Health Research Centre of Oman, No. 392/2014)”.

Materials and Methods:

*Line 113 This implies that some individuals from the original populations (WWR-Oman and WWR-UAE) were not mixed. However at Line 90 the authors suggest that all animals were potentially mixed. Please clarify.

The sentence was modified to avoid confusion:

“The breeding program focused on increasing population size, rather than on increasing genetic diversity, therefore individuals from the two source populations (WWR-Oman and WWR-UAE) were

allowed to mate randomly. Offspring of these source populations are referred in this paper as the WWR-Mix group or herd”.

*Throughout the results the authors refer to five groups of oryx: Oman founder, USA founder, WWR Mix, WWR Oman and WWR UAE. However it is not clear from Table 1 where the USA founder animals are, as none appear to have been analysed using RAD-seq. This needs clarifying. Please make Table 1 consistent with other Figures and Tables and explain clearly at the start of the Methods, where it states that 3 groups were used (Line 113).

Table 1 was modified following reviewer’s comments.

Table 1. Details of samples from the Historical ‘World Herd’ and the current WWR herd used for this study.

Group	Reserve/Zoo	Samples used for Mt-DNA analyses	Samples used for ddRAD analyses	Description
UK founder*	London	1	-	A male born in London RP (London Zoo) which originally came from ‘World Herd’ and translocated to KSA in 1989.
KSA founder*	Thumamah	5	-	These were translocated from the ‘World Herd’ in USA to KSA.
USA founder*	SD-WAP	4	1	San Diego Wildlife Safari Park, Escondido, USA.
	San Diego	1	-	San Diego Zoo, San Diego, USA.
	Phoenix	2	-	Phoenix Zoo, Phoenix, Arizona, USA.
	Brownsville	1	-	Gladys Porter Zoo, Brownsville, USA.
	Los Angeles	1	1	Arabian oryx ‘World Herd’ Los Angeles Zoo & Botanical Gardens.
Oman Founder*	Jalooni	18	10	Jalooni Reserve Office for Conservation Advisor, Muscat, Oman, Currently Office for Conservation of the Environment. These samples represented the founder of Omani Arabian oryx reintroduction project in 1980s.
UAE founder*	Al-Ain Zoo	3	-	Al-Ain Zoo, Al-Ain, UAE. These animals represent the founders’ of reintroduced oryx of Al-Ain Zoo in UAE.
WWR-Mix	WWR-Mix	108	88	WWR-Mix represent the offspring which results from crossbreeding of Oman-Herd and WWR-UAE which were translocated to the WWR in late 2011.
WWR-Oman	WWR-Oman	13	10	WWR-Oman Herd: represented oryx which retrieved from the wild during 1996 – 2007.
WWR-UAE	WWR-UAE	17	15	WWR-UAE Herd: represent the Arabian oryx introduced to WWR in 2011.
	Total**	174	125	

UK: United Kingdom; KSA: Kingdom of Saudi Arabia; UAE: United Arab Emirates; WWR: Al-Wusta Wildlife Reserve.

***Samples obtained from Marshall et al. (1999).**

****The difference in the number of samples between those use for mtDNA and RAD-seq was due the suitability and quality of DNA for downstream experiments or did not yield results suitable data.**

*Line 158 What is meant by a ‘Bonferroni correction of <0.05’? What was the P-value employed after Bonferroni correction? How many populations were assumed etc.

Information was added to the paragraph and the information is now presented in a clearer way:

“We also discarded loci with significant departure from Hardy-Weinberg proportions, within any one group after Bonferroni correction with a p-value of less than 0.05. To calculate deviation from Hardy-Weinberg proportions, we assumed five groups: USA founder, Oman founder, WWR Mix, WWR Oman and WWR UAE”.

*Line 179 ‘...rely on...’

Grammatical mistake was corrected.

*Line 198 It is not clear how an estimate of N_e would inform how different groups have been affected by drift / inbreeding, unless you assume the three (or five?) groups had identical starting conditions. Please explain.

We added a more detailed definition of effective population size to make clear our goal to estimate this parameter.

“The effective population size (N_e) is related to genetic drift and thus predicts rates of loss of neutral genetic variation, fixation of deleterious and favourable alleles and the increase in inbreeding experienced by a population (England et al. 2006). N_e was estimated using the linkage disequilibrium method of Waples and Do (45) as implemented in the software NeEstimator V2.1 (46)”.

Results:

*Line 235 Please clarify whether this is considered to be 58% of extant diversity, or 58% of total diversity ever observed?

In this sentence we referred to the extant diversity, the sentence now reads:

“...the WWR as a whole contains 58% of the extant mtDNA haplotype diversity.”

*Line 241 Reference error

Reference was corrected:

“Within cluster I (Fig. 3), three subclusters (A/B/C, G/F/H and D/E/J) appear to be formed”.

*Line 268 onwards Description of PCA results appears confused. PC2 differentiates WWR-UAE , not WWR-Oman as far as I can tell.

We thanked to the reviewer to bring this mistake into our attention, now the sentence reads:

“PC2 explained 4.9% of the variance in the data and separated the WWR-UAE individuals from the WWR-Mix individuals (Fig. 5)”.

*Line 284 F_{ST} values. Some of these are actually quite high (>12%) and they vary a lot: <1% to >12%. Have they been tested for significance? More accurate description pairwise F_{ST} data required for Table 3.

We agree with the reviewer opinion, so we deleted the words: “relatively low values of variation between populations”. This sentence now reads: “AMOVA analyses and F_{ST} values were significantly

different from zero (Table 3; Supplementary Table 3) indicating that the main source of variation is mostly within groups rather than between groups”.

Additionally, we add the following information in the methods section: “confidence intervals and p-values were generated by bootstrapping across loci”.

Improved the information in Table 3 as follows:

Table 3. Pairwise genetic differentiation (F_{ST}) following Weir & Cockerham¹¹. confidence intervals (CI) and p-values were generated by bootstrapping across loci using the R package dartR (36).

Group 1	Group 2	F_{ST}	Lower bound CI limit (95%)	Upper bound CI limit (95%)	p-value
USA founder	Oman founder	0.082	0.063	0.100	>0.01
USA founder	WWR Mix	0.063	0.047	0.077	>0.01
USA founder	WWR Oman	0.091	0.067	0.108	>0.01
USA founder	WWR UAE	0.121	0.098	0.143	>0.01
Oman founder	WWR Mix	0.027	0.023	0.034	>0.01
Oman founder	WWR Oman	0.008	0.001	0.013	>0.01
Oman founder	WWR UAE	0.121	0.109	0.133	>0.01
WWR Mix	WWR Oman	0.010	0.005	0.013	>0.01
WWR Mix	WWR UAE	0.054	0.048	0.060	>0.01
WWR Oman	WWR UAE	0.113	0.100	0.128	>0.01

*Line 288 Replace ‘parentage’ with ‘familial’

Word was replaced as suggested.

*Line 294 Commentary on the usefulness of a group for conservation should be moved to the Discussion. Also, consider that the UAE, although showing greater inbreeding, may also carry distinct genetic diversity.

This sentence was moved to the discussion section as suggested.

Discussion:

*General: The finding of a distinct mtDNA group is one of the most interesting findings of the work and is probably worthy of more discussion. Do the authors recommend increasing the frequency of these haplotypes in the Oman population. Are there any historical (phylogeographic) explanations for this division? Does it mirror any other patterns of differentiation in Arabian fauna?

We did a revision of the literature looking for the patterns that the reviewer suggested. However, we could not find any relevant study showing patterns of genetic differentiation in the Arabian peninsula.

We mentioned the importance of increasing the frequency of the distinct mt DNA group in the discussion, as follows:

“Particularly important for retaining high levels of genetic diversity would be to prioritize the reproduction of individuals carrying low frequency haplotypes such as Haplotype E, which was identified in a single individual.”

*Line 313 Would replace ‘confirms’ with ‘indicates’.

Word was replaced as suggested.

*Line 325 How do the three groups in the nuclear data compare to the three management herds in Oman? Could the results be due to drift based on isolation of these three herds? If not, is the observed pattern of diversity expected to persist under current management? This is important and more Discussion is required here

This important information was further discussed and added at the end of the paragraph. Additionally, a figure was added in the supplementary information, which shows that genetic patterns are not determined by the three management herds. This paragraph now reads:

“Additionally, we observed that the three genetic ancestries that were identified are randomly distributed among the three management herds (enclosures) at WWR (Supplementary Fig. 8). This observation indicates that genetic patterns presented in this study were not due to drift based on isolation of these three herds. We expect that this random pattern persists under the current management, which is based on random rotation of individuals between enclosures.”

Supplementary Figure 8. Principal component analysis (PCA) using the first two principal components showing that clusters of individuals are not determined by the three management herds (enclosures).

*Line 377 Sentence doesn't make sense

Sentence was removed.

Figures and Tables:

*Figure 1. The Legend is extremely long. Recommend reducing if possible, although the information here is valuable - maybe include long version in Supp. Mat..

The legend was shortened and the extra information was moved to the Supplementary material. The legend now reads:

“Timeline of the Arabian oryx reintroduction program in Oman and establishment of the “World Herd”. Top arrows display the years when the animals were captured, translocated or released and the country where the event took place. Middle boxes show the major event and activity at that time. Bottom boxes show the number and sex of animals which had been used (Stanley-Price, 1989; Marshall et al., 1999; Spalton et al., 1999). Additional details can be found at the Supplementary material.”

The information in the supplementary material reads:

“For the establishment of the “World Herd” two additional females from Kuwait were added but died without contributing to the program. Oman established the reintroduction program at the WWR (also referred to as Jalooni, Jaaluni or Yalooni) with the animals received from the “World Herd” in 1980s. The reintroduction program started with the release of four males and four females in 1982 (Stanley-Price, 1989). A subsequent release consisted of 10 males and 10 females between 1982 and 1995 which were also from the “World Herd” collection in the USA. This introduced population was further enriched with five animals from Jordan and nine from the Oman collection at Bait Al Barakah Royal farm (Omani Mammals Breeding Centre) which both originated from the “World Herd” (Stanley-Price, 1989; Marshall et al., 1999). Pedigree data from the international studbook (Dolan & Sausman, 1992) suggested that those Arabian oryx at Jalooni in 1989 were derived from 13 wild founders caught from different geographical origins including Oman, Yemen, Bahrain and the Kingdom of Saudi Arabia (KSA). Half of these founders originated from the latter country, specifically, the Old Riyadh population (Mace, 1989).”

*The Legend states that the middle boxes show conservation status, but I cannot see this.

The conservation status was removed from the legend, now figure and legend agree.

Figure 2

*Suggest straightening and thinning the black line. Its only needs to join the edges of the relevant areas, not the middle. The maps apart o have been copied from elsewhere. Include information on the source of the original maps. The right hand map contains place names that are too small to read. Suggest either removing or making bigger.

Figure 2 was improved as suggested. Now the figure shows Jalooni in an inset, names that were too small were removed and the map is now correctly attributed as follows:

“Map of the Arabian Peninsula created by Lokal Profil and licensed under CC-BY-SA-2.5.”

Figure 3

*Legend box and right hand cluster box are missing sides

We did not observe the missing sides in the figure as noted by reviewer 1, this might be due to picture format which might be displayed differently in different operative systems. To solve this issue we converted the image to a different format (TIFF).

*Again, the legend text is too long, suggest reducing and placing extra text in Supp. Mat.

The legend was shortened and now reads:

“Median Joining Network (MJN) showing the relationships and clustering of 12 mtDNA CR haplotypes (A through L) including those from published studies. Circle sizes are proportional to the haplotype frequencies, and the length of the lines corresponds to the number of mutations connecting the haplotypes except for the branch leading from the median vectors (black small circles) that joins clusters I and II, which represent seven mutations. Horizontal small lines on branches represent the mutational steps between haplotypes. Further details on the distribution of the mtDNA haplotypes

found among the other captive and reintroduction programs in the Arabian Peninsula is provided in Supplementary Table 1.”

Figure 4

*Final sentence of the legend refers to identification numbers of individuals on the x-axis, but these are impossible to read and are not critically important - suggest removing.

Final sentence and individual labels from figure were removed as suggested:

Figure 4. Bayesian clustering analysis based on 1,091 loci of the five Arabian oryx groups (Oman founder, USA founder, WWR-Oman, WWR-UAE and WWR-Mix) using the software fastSTRUCTURE (Raj et al 2014). Individuals are shown as vertical bars coloured in proportion to their estimated ancestry within each of the inferred populations ($K=3$).

*Table 1 Please align the sample categories with the five ‘groups’ used throughout the results.

Table 1 was modified following reviewer’s comments.

Table 1. Details of samples from the Historical ‘World Herd’ and the current WWR herd used for this study.

Group	Reserve/Zoo	Samples used for Mt-DNA analyses	Samples used for ddRAD analyses	Description
UK founder*	London	1	-	A male born in London RP (London Zoo) which originally came from ‘World Herd’ and translocated to KSA in 1989.
KSA founder*	Thumamah	5	-	These were translocated from the ‘World Herd’ in USA to KSA.
USA founder*	SD-WAP	4	1	San Diego Wildlife Safari Park, Escondido, USA.
	San Diego	1	-	San Diego Zoo, San Diego, USA.
	Phoenix	2	-	Phoenix Zoo, Phoenix, Arizona, USA.
	Brownsville	1	-	Gladys Porter Zoo, Brownsville, USA.
	Los Angeles	1	1	Arabian oryx ‘World Herd’ Los Angeles Zoo & Botanical Gardens.
Oman Founder*	Jalooni	18	10	Jalooni Reserve Office for Conservation Advisor, Muscat, Oman, Currently Office for Conservation of the Environment. These samples represented the founder of Omani Arabian oryx reintroduction project in 1980s.
UAE founder*	Al-Ain Zoo	3	-	Al-Ain Zoo, Al-Ain, UAE. These animals represent the founders’ of reintroduced oryx of Al-Ain Zoo in UAE.

WWR-Mix	WWR-Mix	108	88	WWR-Mix represent the offspring which results from crossbreeding of Oman-Herd and WWR-UAE which were translocated to the WWR in late 2011.
WWR-Oman	WWR-Oman	13	10	WWR-Oman Herd: represented oryx which retrieved from the wild during 1996 – 2007.
WWR-UAE	WWR-UAE	17	15	WWR-UAE Herd: represent the Arabian oryx introduced to WWR in 2011.
	Total**	174	125	

UK: United Kingdom; KSA: Kingdom of Saudi Arabia; UAE: United Arab Emirates; WWR: Al-Wusta Wildlife Reserve.

***Samples obtained from Marshall et al. (1999).**

****The difference in the number of samples between those use for mtDNA and RAD-seq was due the suitability and quality of DNA for downstream experiments or did not yield results suitable data.**

*Supp Table 3 Column headers (Sigma, Percentage and Phi) need more explanation. What does this table show?

The following text was added:

“Sigma is the variance; Percentage is the percent of the total variance explained by each source of variance; Phi is the population differentiation statistic. A higher Phi value represents a higher amount of differentiation.”

*Supp Table 4 Explain F_h and F_{alt} in the legend

The following text was added:

“Inbreeding was estimated using two different statistics: F_h which is a deviation in homozygosity from its Hardy–Weinberg expectation using the software PLINK (Purcell *et al.*, 2007); and, F_{alt} where homozygous loci are weighted with the inverse of their allele frequency using the software GCTA (Yang *et al.*, 2011).”

*Supp Table 7 How do you define ‘highest’?

The text was modified to avoid confusion and now reads:

“Inbreeding estimates (F_h and F_{alt}) by individual, ordered from the most inbred to the least inbred.”

Reviewer: 2

Comments to the Author(s)

General comments

Using mitochondrial sequences and SNP genotypes generated from RAD-sequencing, the authors analyzed the genetic diversity, inbreeding, structure, and kinship among Arabian oryx maintained at the Al-Wusta Wildlife Reserve in Oman, where an important captive breeding and reintroduction program was established for this species, and among a set of historical samples collected in the 1980s, when the first captive breeding programs for the species were initiated. The animals at the Al-Wusta Wildlife Reserve originate from different sources. Therefore, understanding the genetic contributions and consequences of these sources are useful for the

conservation breeding and reintroduction programs, which have lacked formal genetic management strategies until now. The data and results the authors have generated represent an important advancement in our understanding of the genetics of Arabian oryx captive populations and also represents a nice model of how genomic data can be directly implemented to inform breeding recommendations (in the immediate future) and reintroductions (a more distant goal).

The manuscript is generally well-written and presents entirely novel data and results. The amount of text devoted to the introduction, materials and methods, results, and discussion sections is well balanced. The cited literature is comprehensive and up to date. The mitochondrial and RAD-seq data have been analyzed in a comprehensive and rigorous manner, and the authors have appeared to pay attention to sample size issues and the underlying assumptions of the analytical methods they applied. I was especially intrigued with the use of the q-profile approach developed for evolutionary and population genetic applications by Sherwin et al. 2017. This approach standardizes estimates of population genetic parameters into effective measures that make them easily comparable across different sampled groups. The figures and tables are generally of good quality and easily interpretable. The additional results presented in the supplementary figures and tables provide important details that support the conclusions. I especially like Supplementary Figure 6 (the larger heatmap of IBD among all sampled individuals). The manuscript suffers from quite a number of grammatical errors that diminish clarity and readability. I provide comments and recommendations regarding these and other issues for the authors to address, which I think would help improve this important manuscript.

Specific comments

*1. Lines 47-48: "IUCN red list of threatened species" This is a formal name and therefore the full name should be capitalized: "IUCN Red List of Threatened Species"

The name is now capitalized:

"...the IUCN Red List of Threatened Species"

*2. Lines 58-59: Sentence awkwardly worded: "The current number of the Arabian oryx is estimated ~1,100 in the wild and ~6,000-7,000 in captivity, zoos, reserves and private collections (1)." Suggested revision: "The current number of the Arabian oryx is estimated to be ~1,100 in the wild and ~6,000-7,000 in captivity, which includes zoos, reserves and private collections (1)."

The sentence was revised as suggested.

*3. Lines 78-79: "Eventually, through successful management, the Jalooni population increased to about 400 captive and reintroduced individuals into the wild by 1996 (16)." As written, it's not clear whether the authors' mean a total of 400 individuals from both captive and wild populations were present in 1996 or that by 1996, 400 individuals had been reintroduced into the wild. I assume the former was the intent of the sentence. In that case, remove the phrase "into the wild" to clarify the meaning of the sentence.

The sentence was modified to avoid confusion:

"Eventually, through successful management, the Jalooni population increased to about 400 individuals by 1996, which included individuals in captivity and individuals released into the wild."

*4. Line 97: "World Herd" is misspelled in this line.

Misspelled word was corrected.

*5. Line 113: "...its primary source population..." As this phrase refers to the "samples" (plural), this should be changed to "...their primary source population..."

Grammatical mistake was corrected.

*6. Line 114: Add a 'the' before "result" in this sentence.

The word "the" was added.

*7. Line 122: For the sake of readers, specify the length of the mtDNA CR sequence that was amplified. This is indicated as 638 base pairs in the Results section. However, it's is useful to mention this here in the Methods section.

The number of base pairs were added as suggested:

"MtDNA CR sequences (638 base pairs) were amplified..."

* 8. Lines 122 and 134: "MtDNA CR" and "mtDNA CR" To improve clarity, these should be qualified by the word 'sequences': "MtDNA CR sequences" and "mtDNA CR sequences"

The word "sequences" were added in all the cases.

*9. Line 131: Specify the method and instrument used to quantify libraries. I assume using a qPCR?

Information about library quantification was added:

"Libraries were quantified fluorimetrically with a Qubit 2.0 (Thermo Fisher Scientific)."

*10. Line 152: "...to obtain equal length of sequences..." Delete the "of" in this sentence.

The word "of" was deleted.

*11. Lines 154-155: What to the authors mean by "individuals sampled more than once"? Were some individuals sequenced more than once in the RAD-seq protocol? If so, why and what proportion of the 135 samples used for the RAD-seq analyses?

Four samples were sequenced two times for quality control. We deleted this sentence to avoid confusion.

*12. Line 172: "explain" should be changed to "explains" in this sentence.

Grammatical mistake was corrected.

*13. Line 179: "PCA is a technique that does not relay in any genetic assumption..." I assume the authors meant ""PCA is a technique that does not rely on any genetic assumption..."?

Grammatical mistake was corrected.

*14. Line 187: "...to be used in samples of unequal sample sizes..." To improve clarity, change to "...to be used among groups of unequal sample sizes..."

Sentence was modified as suggested.

*15. Lines 189-190: "...to allow a direct comparison." A comparison of what, exactly? Among groups? Populations? Please specify.

Sentence was modified to avoid confusion:

"These measures are converted to a common scale of effective numbers (Hill's numbers) to allow a direct comparison between these measures".

*16. Line 191: Add a "the" before "Shannon index."

The word "the" was added.

*17. Line 192: Add a "a" before "measure."

The word "a" was added.

*18. Lines 194-195: "To identify the group contributing with most to the variance of genetic variation..." Suggested revision: "To identify the groupings contributing the most to the variance of genetic variation..."

Sentence was modified as suggested.

*19. Line 216: Delete "Yang et al." in this sentence.

"Yang et al." was deleted.

*20. Line 221: "MtDNA control region (638 bp) was generated..." To improve clarity and grammar, change to: "MtDNA control region sequences 638 bp in length were generated..."

Sentence was modified as suggested.

*21. Line 225: "Haplotypes sequences are available in GenBank..." Suggested revision: "Haplotype sequences are available from GenBank..."

Sentence was modified as suggested.

*22. Lines 238-239: In Figure 3: these two haplogroups are referred to as Cluster I and Cluster II. For the sake of consistency, the authors should select one of these - haplogroup or cluster.

We used the word "cluster" throughout the text.

*23. Line 239: I think the authors mean haplogroup (or Cluster) I here.

We thanked to the reviewer to bring this mistake into our attention, now the sentence reads:

“Within cluster I (Fig. 3), three subclusters (A/B/C, G/F/H and D/E/J) appear to be formed”.

*24. Line 252: Change “no present” to “not present” in this sentence.

Sentence was modified as suggested.

*25. Line 254: Add “was” before “comprised” in this sentence.

The word “was” was added.

*26. Line 255: Add a “the” before “National Center for Biotechnology Information”

The word “the” was added.

*27. Line 259: Awkward wording: “The inferred number of populations (K) that explained the best the genetic structure...” Suggested revision: “The inferred number of populations (K) that best explained the genetic structure...”

Sentence was modified as suggested.

*28. Line 262: Add a comma after “likelihood” in this sentence.

A comma was added as suggested.

*29. Line 272: “founders/descendent.” To improve clarity, this should be changed to “founders and descendents”

Sentence was modified as suggested.

*30. Line 298: Change “less inbred” to “least inbred”

Sentence was modified as suggested.

*31. Line 300: Remove the comma after “PCA” in this sentence.

The comma was removed as suggested.

*32. Lines 314-315: Awkward sentence construction: “The additional four haplotypes were identified in the WWR individuals sourced from the Arabian Peninsula including KSA and UAE...” Suggested revision: “The additional four haplotypes identified in the WWR individuals were sourced from the Arabian Peninsula, including KSA and UAE...”

Sentence was modified as suggested.

*33. Line 333: Change “recently” to “recent”

Sentence was modified as suggested.

*34. Line 343: Add a comma after “group” in this sentence.

A comma was added as suggested.

*35. Lines 351-352: Awkward sentence construction: “After, acknowledging the above caveats, overall our results showed...” Suggested revision: “After acknowledging the above caveats, our results showed...”

Sentence was modified as suggested.

*36. Line 366: it’s not clear what the authors mean about $q=0$ having “sampling problems.” Do they mean that the estimation $q=0$ is influenced by sample size of populations measured? Please revise to specify the problems associated with the q -profile approach.

The sentence was modified to avoid confusion:

“For instance, $q=0$ is sensitive to rare alleles (i.e. alleles present in low frequencies in the population) but its estimates are highly dependent on sampling size, whereas $q=2$ shows the opposite trend (38)”.

*37. Add a comma after “Together” in this sentence.

A comma was added as suggested.

*38. Line 382: Change “by” to “in” – “in the USA founder group.”

Grammatical mistake was corrected.

*39. Line 386: Change “use” to “used” in this sentence.

Grammatical mistake was corrected.

*40. Lines 396-397: “...aimed to maximize genetic diversity and control the negative effects of inbreeding and evolutionary potential.” How is controlling evolutionary potential a negative effect? I think the authors meant to write: “...aimed to maximize genetic diversity and evolutionary potential and control the negative effects of inbreeding.”

The sentence was modified to avoid confusion:

“By using the genetic data generated in this study, the breeding program at the WWR could move from a random breeding approach to one with a clear strategy aimed to maximize genetic diversity and evolutionary potential and control the negative effects of inbreeding (60)”.

*41. Line 413: Add a comma after “Arabian Peninsula” in this sentence.

A comma was added as suggested.

*42. Line 415: “long” should be changed to “long-term” in this sentence.

Sentence was modified as suggested.

*43. Line 417: “separating offspring” – the authors need to indicate when such a separation would occur. I assume after weaning? Or when individuals reach sexual maturity. Such a recommendation has important implications and therefore needs to be better specified.

Information about separating offspring has been added:

“separating offspring (before reaching sexual maturity, *i.e.* 1-1.5 years)”

*44. Lines 421-424: Awkward sentence construction: “...establishing satellite breeding groups, this could be smaller breeding nucleus with multiple females complemented with a seasonal rotational male breeding scheme so females could had the opportunity to breed with other representatives males (63) from the genetic lineages identified in this study;” Suggested revision: “...establishing satellite breeding groups, which would include a smaller breeding nucleus with multiple females, complemented with a seasonal rotational male breeding scheme so that females have the opportunity to breed with other representative males (63) from the genetic lineages identified in this study;”

Sentence was modified as suggested.

*45. Lines 429-430: “from a continuing genetic monitoring program” Do the authors mean using the same genetic approaches as reported in their paper? Please specify, as this will provide useful guidance to all managers of Arabian oryx breeding and reintroduction programs.

We were more specific about the monitoring program. We recommend using similar approaches as the employed in our study to provide a continuity of our results.

“Importantly, the implementation of a genetic-based management approach for the Arabian oryx at WWR would benefit greatly from a continuing genetic monitoring program, using similar methods as those described in this study, to assess and minimise the major threats to the survival of this important reintroduction program.”

*46. Line 436: Change “samples” to “sample” in this sentence.

Grammatical mistake was corrected.

47. Figure 1: Add the word “program” after “reintroduction” in the first sentence of the caption.

The word “program” was added as suggested.

*For the middle boxes of the figure – first middle box: “Conducted of “Operation Oryx” should be revised to “Operation Oryx” conducted”. In the fourth middle box, text can be changed to “Reintroduction of “Oman-Herd into the wild.” In the bottom boxes, “Operation Oryx” in the third box should be capitalized for the sake of consistency. Similarly, “World-herd” can be changed to “World Herd.” in the fourth box.

Figure was improved and suggestions were incorporated.

*48. Figure 2: Awkward wording: "Jalooni located in west north of Al-Wusta Wildlife Reserve." Suggested revision: "Jalooni is located in northwestern part of the Al-Wusta Wildlife Reserve."

Sentence was modified as suggested, and now reads:

"Jalooni is located in the north-west part of the Al-Wusta Wildlife Reserve".

*Also, please specify what the purple blotches indicate in the map figure on the left. I assume these are the Arabian oryx reserves in Israel, Kuwait, KSA, Oman and UAE?

The figure was improved and the purple blotches were removed. The final figure is as follows:

*49. Figure 3: Change the period to a comma before "Jordan and USA (KU985184, Ochoa et al., 2016)" in the caption. Also, add a comma before "with seven haplotypes (A-G)..." In addition, revise the following sentence: "Further details on the distribution of these haplotypes among other the captive and reintroduction programs"

in the Arabian Peninsula...” as “Further details on the distribution of these haplotypes among the other captive and reintroduction programs in the Arabian Peninsula...”

Addition of commas, correction of grammatical mistakes and suggested revisions were done as suggested.

*50. Figure 4: The clustering plot and individual ID numbers at the bottom need to be increased in size to improve readability. In the current version, the IDs are too small to be read.

Increasing the size of the individual labels resulted in labels overlapping, therefore we decided to remove these labels:

Figure 4. Bayesian clustering analysis based on 1,091 loci of the five Arabian oryx groups (Oman founder, USA founder, WWR-Oman, WWR-UAE and WWR-Mix) using the software fastSTRUCTURE (Raj et al 2014). Individuals are shown as vertical bars coloured in proportion to their estimated ancestry within each of the inferred populations (K=3).

*51. Figure 7 caption: “Endelman & Jannink19” should be changed to “Endelman & Jannink (2012)”

Reference was modified as suggested.

*52. Table 1: Are specific years in the 1980s available for the historical samples?

We added a table in the supplementary information containing the years for the historical samples.

Supplementary Table 2. General information of the groups analysed in this study.

	Country of Origin	Country End Point	Sex		Year	Source
			Male	Female		
Historical Samples (World Herd)	SD-WAP USA	Yalooni OMAN	1	2	1977-1982	Marshall et al., 1999
	Los Angeles USA	Yalooni OMAN	0	1	1979	Marshall et al., 1999
	Yalooni Oman	Yalooni OMAN	9	4	1980-1990	Marshall et al., 1999
	Thumamah KSA	Thumamah KSA	1	2	1987	Marshall et al., 1999
	AL Ain UAE	NWRC KSA	2	1	1990	Marshall et al., 1999
	Phoenix USA	Yalooni Oman	1	0	1971	Marshall et al., 1999
Current WWR	UAE	UAE	3	14	< 2009	This study
	Oman-Mix	Oman	38	49	2010-2015	This study
	Oman-herd	Oman	1	9	< 2009	This study

*53. Tables 2 and 3: Can the authors test for significant difference in the values (e.g., Shannon index, Fst) among the groups reported in these tables?

Standard deviations and significance were added in tables 2 and 3.

Table 2. Summary statistics calculated in each group. Standard deviation is shown in parentheses.

Group	Number of individuals	Allelic richness	Shannon index	Observed heterozygosity	Expected heterozygosity	Inbreeding coefficient	Ne	Ne CI Low - CI high
Oman founder	10	1.268 (0.194)	0.386 (0.255)	0.263 (0.223)	0.268 (0.194)	0.023 (0.312)	16	15.2-16.8
USA founder	2	1.259 (0.281)	0.282 (0.301)	0.257 (0.325)	0.259 (0.278)	0.007 (0.371)	NA	NA
WWR Mix	89	1.281 (0.166)	0.431 (0.211)	0.274 (0.170)	0.281 (0.166)	0.022 (0.136)	22.4	22.2-22.7
WWR Oman	9	1.270 (0.195)	0.387 (0.255)	0.281 (0.235)	0.270 (0.195)	-0.027 (0.288)	44.3	39.4-50.4
WWR UAE	15	1.256 (0.193)	0.374 (0.261)	0.233 (0.199)	0.256 (0.193)	0.072 (0.292)	2.1	2-2.1

The lower the inbreeding coefficient values the lower the level of inbreeding.

Ci - confidence intervals. Ne in the USA founder group was not calculated due to the low number of individuals sampled) n=2.(

Table 3. Pairwise genetic differentiation (F_{ST}) following Weir & Cockerham¹¹. confidence intervals (CI) and p-values were generated by bootstrapping across loci using the R package dartR (36).

Group 1	Group 2	F_{ST}	Lower bound CI limit (95%)	Upper bound CI limit (95%)	p-value
USA founder	Oman founder	0.082	0.063	0.100	>0.01
USA founder	WWR Mix	0.063	0.047	0.077	>0.01
USA founder	WWR Oman	0.091	0.067	0.108	>0.01
USA founder	WWR UAE	0.121	0.098	0.143	>0.01
Oman founder	WWR Mix	0.027	0.023	0.034	>0.01
Oman founder	WWR Oman	0.008	0.001	0.013	>0.01
Oman founder	WWR UAE	0.121	0.109	0.133	>0.01
WWR Mix	WWR Oman	0.010	0.005	0.013	>0.01
WWR Mix	WWR UAE	0.054	0.048	0.060	>0.01
WWR Oman	WWR UAE	0.113	0.100	0.128	>0.01

Reviewer: 3

Overall comments to authors:

The aim of this paper is to use population genetic tools, using data obtained from one mitochondrial gene and genome-wide SNP genotyping, to describe the levels of genetic diversity and genetic management history of the Arabian oryx in Oman. The authors use a number of analytical techniques to calculate multiple population genetic measures and provide implications for future genetic management based on the results of past management.

This is a well-written paper, with appropriate references and all methods and results are easy to read and interpret. The authors are to be commended on this submission, which will add to the growing body of literature using molecular genetic and empirical data to manage ex situ populations.

*The main suggestion is for the authors to expand the discussion, particularly in the management implications suggestion, to be more specific in their recommendations for genetic management. They should spend some

time discussing both the feasibility of their recommendations, but also the differences and trade-offs in future genetic implications when applying the individual genetic management recommendations compared to more group or herd-level recommendations. The authors make suggestions that are more intensive and would require more intensive data collection and pedigree tracking and are more useful in an individual-management approach, versus those that are less intensive such as rotating breeding males, which are somewhat less intensive. This part of the discussion felt more like a list of options, rather than considering their results to make specific recommendations and considering the genetic implications of each recommendation.

We used the valuable reviewer comments about how to improve the discussion.

Section by Section comments:

Abstract

*Line 25: It would be more clear on the dates of events to explicitly state that the reintroduction commenced a decade after the species was declared extinct in the wild (1982 specifically, or 1980 as in Figure 1).

The sentence was modified as suggested:

“The reintroduction of this species into the wild commenced a decade later after the species was declared extinct in the wild from two main sources...”

Introduction

*Line 63: Instead of simply “captive breeding was not managed through studbooks”, expand upon this to state something to the effect of “captive breeding was not implemented using typical individual genetic management through minimizing population mean kinship”. The studbook is the database to maintain data, but is not the tool for management.

Sentence was modified as suggested:

“In some cases management details were not recorded (or published) and captive breeding was not implemented using typical individual genetic management through minimizing population mean kinship”.

*Lines 64-70: It would be helpful to include more specific genetic statistics from these papers to give the reader an idea of what is meant by “low levels of genetic diversity” and “high levels of inbreeding and relatedness” relative to the “high level of genetic diversity” found in the Saudi Arabian group.

We added a table describing the genetic statistics in the Arabian Oryx papers and the table was referenced accordingly in the lines mentioned by the reviewer.

Supplementary table 1. Summary of genetic statistics found in Arabian Oryx papers.									
Source	Group	Country	No. individuals	No. markers	Molecular markers	H_o	H_e	F_{IS}	mtDNA haplotypes
Eljara et al 2017	Wadi Rum protected area	Jordan	49	11	Microsatellites	0.535	0.435	-0.230	-
Arif et al 2010	Mahazat as-Sayd protected area and	Saudi Arabia	24	7	Microsatellites	0.601	0.565	-0.064	-

	National Wildlife Research Centre, Taif								
Elmeer et al. 2019	Mas-Habiyya	Qatar	23	13	Microsatellites	0.332	0.583	0.431	-
Elmeer et al. 2019	Ras Usheirij	Qatar	7	13	Microsatellites	0.388	0.46	0.157	-
Elmeer et al. 2019	Private farm A	Qatar	14	13	Microsatellites	0.416	0.529	0.214	-
Elmeer et al. 2019	Private farm B	Qatar	10	13	Microsatellites	0.388	0.631	0.385	-
Elmeer et al. 2019	Al Wajba	Qatar	14	13	Microsatellites	0.422	0.455	0.073	-
Elmeer et al. 2019	Shahaniya	Qatar	14	13	Microsatellites	0.385	0.49	0.214	-
Elmeer et al. 2019	Doha Zoo	Qatar	14	13	Microsatellites	0.436	0.51	0.145	-
El Alqamy et al. 2011	Sir Bani Yas Island	UAE	38	13	Microsatellites and mtDNA	-	0.337		3
El Alqamy et al. 2011	Al Bahia	UAE	20	13	Microsatellites and mtDNA	-	0.384		5
El Alqamy et al. 2011	Al Ain Zoo	UAE	32	13	Microsatellites and mtDNA	-	0.341		1
Khan et al. 2010	King Khalid Wildlife Research Centre, Thumamah and Mahazat as-Sayd protected area	Saudi Arabia	24	-	mtDNA	-			7
Marshall et al. 1999	SDWAP	USA	90	6	Microsatellites	0.496	0.509	0.026	-
Marshall et al. 1999	Arabian Oryx Sanctuary	OMAN	77	6	Microsatellites	0.43	0.47	0.085	-
Marshall et al. 1999	King Khalid Wildlife Research Centre, Thumamah,	Saudi Arabia	34	6	Microsatellites	0.624	0.626	0.003	-
Marshall et al. 1999	National Wildlife Research Centre, Taif	Saudi Arabia	97	6	Microsatellites	0.622	0.575	-0.082	-

Ho - observed heterozygosity; *He* - expected heterozygosity; *F_{IS}* - inbreeding coefficient calculated as 1-(*Ho/He*).

*Lines 72-73: Could the authors provide the captive populations that were sampled for these mtDNA studies to allow the reader to understand where the 12 haplotypes were identified?

The following table was added in the supplementary information and referenced accordingly.

	Country of Origin	Country End Point	Sex		Year	Source
			Male	Female		
Historical Samples (World Herd)	SD-WAP USA	Yalooni OMAN	1	2	1977-1982	Marshall et al., 1999
	Los Angeles USA	Yalooni OMAN	0	1	1979	Marshall et al., 1999
	Yalooni Oman	Yalooni OMAN	9	4	1980-1990	Marshall et al., 1999
	Thumamah KSA	Thumamah KSA	1	2	1987	Marshall et al., 1999
	AL Ain UAE	NWRC KSA	2	1	1990	Marshall et al., 1999
	Phoenix USA	Yalooni Oman	1	0	1971	Marshall et al., 1999
Current WWR	UAE	UAE	3	14	< 2009	This study
	Oman-Mix	Oman	38	49	2010-2015	This study
	Oman-herd	Oman	1	9	< 2009	This study

*Lines 73-75: It would help to expand upon this statement that the data provided in the previous studies could not be used for intensive individual-based management or individual breeding strategies. The information provided at the population or herd level in previous studies could be used to develop some basic group or herd-management style recommendations.

Following reviewer's comments we improve this statement and now reads:

“The above genetic studies have provided important information to develop basic herd-management recommendations, however they do not have the resolution required to inform intensive individual-based management or individual breeding strategies (7).”

*Line 79: needs clarification. Do you mean the herd grew to 400 individuals and then those individuals were used for reintroduction? Or that on the WWR there were multiple herds delineated by those consisting of captive animals and those containing reintroduced animals? The reviewer believes it is the latter, but more details or explanation in the sentence would help with this and reference to Figure 1 and Table 1 would also clarify this.

Here we referred to the maximum numbers of reintroduced animals achieved in the wild before drop to 70 and returned back to captivity. We rewrote the sentence, as follows:

“Eventually, through successful management, the Jalooni population increased to about 400 individuals in the wild by 1996.”

*Line 94: instead of “breeding strategy”, suggest change to “genetic management strategy”.

Sentence was modified as suggested.

*Lines 100-101: Rephrase, as random mating is one type of management that can work to maintain genetic diversity (and is often a strategy that is better than non-random management without genetic or pedigree data). Likely, the authors intention is to discuss mean kinship based management, which requires a mostly known pedigree and is an individual-based management strategy. Group management can also be done via group or herd kinship, but this specific group management strategy does not appear to be the aim of the authors and the reason for collecting these genetic data.

Random mating, as currently performed at WWR, is a type of management that can maintain adequate levels of genetic diversity. However, the acquisition of genetic data could allow the transition to a strategy based on the management of individuals or groups to maximise genetic diversity.

Materials and Methods

*Line 110-111: Please check reference/citation.

Reference was corrected:

Blood samples were obtained from 138 randomly selected individuals at WWR (Fig. 2; Animal Welfare Committee at the Animal Health Research Centre of Oman, No. 392/2014).

Lines 112-114: Please provide more details as to how samples were known to be from each lineage. The introduction states that no studbook or records were maintained, but some information was available to do this pre-sorting into an assumed lineage.

This information was added to the text, as follows:

“Individuals were identified by ear tags, which include the source group.”

*Line 179: Should this say “rely on” instead of “relay in”?

Grammatical mistake was corrected.

Results

*Line 239: Should this be “within haplotype I”?

We thanked to the reviewer to bring this mistake into our attention, now the sentence reads:

“Within cluster I (Fig. 3), three subclusters (A/B/C, G/F/H and D/E/J) appear to be formed”.

*Line 240 and 271: Please check reference/citation.

Citations and references were corrected.

*Line 252: Correct to “loci NOT present in all groups”

Grammatical mistake was corrected.

*Line 259: extra “the”

The word “the” was deleted as suggested.

Discussion

*Line 333: “recently” should be “recent”

Grammatical mistake was corrected.

Line 347-349: This sentence sets up for the authors to provide a number of reasons that SNP derived data are different from microsatellites, but then the authors provide a single reason. Are there any additional reasons to cite? Line 352 mentions multiple caveats so multiple reasons should be provided.

We added more differences between SNPs and microsatellites as requested by the reviewer:

“For instance, and in general, heterozygosity as measured by SNPs (i.e. based on biallelic loci) tends to be lower than heterozygosity measured by microsatellites (which usually display more than two alleles based on loci with three or more alleles). Furthermore, SNP data provides a more accurate representation of the genome, because current sequencing technologies allow to genotype thousands of markers compared to tens of markers for typical microsatellites datasets.”

*Line 387: The authors state that they provide “molecular and relatedness data” in the current study but it would help to make this sentence more specific to the methods and results and that the authors generated molecular data to calculate X, Y, Z population genetics statistics (not simply relatedness data, which are not explicitly provided in the results).

We re-wrote this sentence based on reviewer's comments and now it reads:

“The current study generated baseline information regarding several measures of genetic diversity and relatedness (e.g. Figs. 3 and 7 and Table 2), which will be beneficial for the future management at WWR. For instance, this information can be used to guide the breeding program including setting defined goals, such as reaching a certain effective population size and specific levels of genetic variation and inbreeding.”

*Line 388-389: It would be helpful for the reader for the authors to expand upon specifically, but briefly, what they mean by “management of this program is informed by genetics with defined goals”. This would strengthen the lead into the subsequent section on management implications.

Defined goals were specifically mentioned:

“Since the current study provides molecular and relatedness data, it will be beneficial if future management of this program is informed by genetics with defined goals, such as reaching a certain effective population size and specific levels of genetic variation and inbreeding.”

*Line 396-397: rephrase sentence structure, right now it reads as if you are suggesting “controlling the negative effects of...evolutionary potential” but likely you mean to suggest to use genetic management to maintain evolutionary potential.

We thanked the reviewer for bringing this mistake into our attention, the sentence now reads:

“By using the genetic data generated in this study, the breeding program at the WWR could move from a random breeding approach to one with a clear strategy aimed to maximize genetic diversity and evolutionary potential and control the negative effects of inbreeding (60)”.

*Line 399-401, 418-420 and the entire section on management implications: Reconstruction of pedigree is one option for future management. Given that the parentage assignment results only uncovered 20 first- and second-order relationships to allow for reconstruction, and the relatedness results were only briefly discussed and not used in the results to develop a system of evaluating individuals' empirical kinships for future pedigree and thus individual management, it may strengthen this paper to use this section to discuss the ability of the managers in these herds to follow through with management recommendations that are more individual based (as recommended in these lines) versus a strategy that uses maximum avoidance of inbreeding, or even more group or herd based methods using the results (Figure 7 as an example) to recommend transfer of individuals into different herds which would improve diversity (as you touched on in the sentence on rotational male breeding). The results in the paper already demonstrate that some of the “random” management strategies have proven successful at maintaining diversity and reducing inbreeding in some herds, so a discussion and separation of the recommendations that are more intensive from a management and data collection perspective, to those that are less intensive would be beneficial. As the ex situ management community aims to demonstrate the benefits and successes of less-intense genetic management strategies, this paper has the data and results to contribute to that discussion and body of literature.

We used the important recommendations of the reviewer to improve our discussion.

Other

*Figure 1: “establishment” spelled incorrectly in caption and some other small typos. Thank you for the inclusion of this timeline and diagram to help orient the reader to the history of the moves and establishment of the different herds and locations.

Grammatical mistakes were corrected.

*Supplementary Table 1: It is a bit unclear the distinction between Historical and Published sources so providing a bit more detail in the table description would help clarify this.

The author information was added to the table.

Supplementary Table 4. Distribution of mtDNA haplotypes among the captive and reintroduction programs in the Arabian Peninsula.															
Country	Dataset source	Sample size	Number of haplotypes	A	B	C	D	E	F	G	H	I	J	K	L
Oman	WWR (this study)	138	7	67	31	15	14	1	7	3	×	×	×	×	×
	Historical (this study)	18	5	12	×	×	×	×	2	1	×	×	×	1	2
KSA	Historical (this study)	5	3	1	2	×	×	2	×	×	×	×	×	×	×
	Khan et al. 2010, Hassanin et al. 2012	27	7	×	4	2	12	6	×	×	1	1	1	×	×
Jordan	Ochoa et al. 2016	21	2	5	×	×	×	×	×	16	×	×	×	×	×
UAE	Historical (this study)	3	1	×	×	×	3	×	×	×	×	×	×	×	×
	El Alqamy et al. 2011	61	7	✓	✓	✓	✓	✓	✓	✓	×	×	×	×	×
UK	Historical (this study)	1	1	✓	×	×	×	×	×	×	×	×	×	×	×
USA	Historical (this study)	9	4	2	×	×	×	3	2	×	×	×	×	2	×
	Ochoa et al. 2016	12	4	✓	×	×	×	✓	✓	×	×	×	×	✓	×

✓Present but the frequency is not available
 ×Not present

Reviewer: 4

Comments to the Author(s)

This manuscript presents a very interesting and worthwhile contribution to the literature on a species that was critically endangered, thought to be extinct in the wild, had its population expanded through managed breeding under human care, was released back into the wild and remains a focal species for private collections. The assessments of genetic diversity previously performed were typically for subpopulations of the overall population, lacked resolving power and made only minor contributions to population management regarding retention of the greatest possible extent of the surviving genetic variation of the species, the Arabian oryx, or white oryx, *Oryx leucoryx*. As the authors state (l 88-90), “The breeding program focused on increasing population size, rather than on increasing genetic diversity.” and (l 99-101), “Additionally, the absence of genetic data has limited the ability of this program to move from a strategy of random mating to a genetic-based (or group-based) management.”

To address the information missing from what has become standard population management practice for conservation of large mammal species, the authors obtained a variety of samples of blood, tissues, or DNA extracts from previous studies and conducted two separate studies. One investigated mitochondrial DNA diversity using PCR amplification and DNA sequencing of 174 individuals, including samples from individuals from the early generations of the breeding efforts. The other used reduced representation library sequencing

methods (Rad-Seq) to interrogate presumptively homologous portions of the *O. leucoryx* genome from 135 individuals in the later generations of the breeding efforts.

Usable results were obtained for the mitochondrial analyses from more samples than for the Rad-Seq analysis, a not unexpected result as the authors explained (l 119), “we kept the samples with high DNA quality only.”

A mitochondrial DNA network was generated and, using comparison between the ‘historical’ samples (from the early generations of captive breeding when a pedigree for the World Herd was being kept), and the samples from the recent generations of animals bred in the Sultanate of Oman the authors show that this population (WWR) as a whole contains 58% of the global mtDNA 236 haplotype diversity. There are some indications of structure in the mitochondrial DNA haplotype map, but the total branch length of the network is relatively small and the authors identify two clusters of haplotypes that are separated by seven mutations.

The results of the Rad-Seq analysis are presented several ways: as a PCA plot, a Bayesian cluster analysis and with a heat-map of estimated relatedness. These data and analyses of *F_{st}* combine to suggest that there was modest genetic differentiation in the groups of individuals that have been incorporated into the current managed population in the Sultanate of Oman. Although admixture has taken place between the combined groups, there is still some structure within the overall Oman population.

*Management goals are alluded to, but the paper might have more impact if concrete examples were given.

We gave concrete goals examples of management:

“The current study generated baseline information regarding several measures of genetic diversity and relatedness (e.g. Figs. 3 and 7 and Table 2), which will be beneficial for the future management at WWR. For instance, this information can be used to guide the breeding program including setting defined goals, such as reaching a certain effective population size and specific levels of genetic variation and inbreeding.”

*Disparate founder contributions can have important impacts in analyses like this one when attempting to manage allele loss. It is hard to follow the relationships between the individuals included in the historical samples, but presumably this can be accomplished using studbook data, which might illuminate relationships between the historical individuals and between historical individuals and the founders of the White Oryx Project in Oman. If the authors could do this, their analyses of kinship could potentially be improved.

Unfortunately, there is not studbook data available for historical samples that could improve our analyses as suggested by the reviewer.

*There is little overall structure in the current WWR population, though evidence of its historical origins is evident. The management of breeding is an interesting challenge in this species. The recommendation to prioritize pairings between individuals of low kinship (low IBD) may not necessarily optimize retention of overall genetic diversity. Inbred individuals can have unique variation, as is suggested in the analysis of the UAE population. Rather, it would be important to manage allelic loss so as to retain as much overall variation as possible. Altogether, the issue of potential impacts on management can be expanded and clarified.

We agree with the reviewer. We added further recommendations based on reviewer’s comments:

“Particularly important for retaining high levels of genetic diversity would be to prioritize the reproduction of individuals carrying low frequency haplotypes such as Haplotype E, which was identified in a single individual.”

*One concern with SNP calling in Rad-Seq is aligning paralogs, which increases heterozygosity and can give rise to inaccurate analyses of relationships and population diversity. The methods could benefit from greater descriptive detail. While perhaps not generally known, the genome of *O. leucoryx* has a large repetitive component that has been encountered when producing and assembly using short-read data. So, the concern about paralogs is appropriate.

We performed analyses to search for paralogs, in which we did not find any pair of sequences that could be paralogs. We included this analysis in methods and results:

Methods:

“Additionally, we discarded loci with sequences that are too similar which might be possible paralogues (a pair of genes that derives from the same ancestral gene and now reside at different locations within the same genome) by using the function `gl.filter.hamming` from the R package `dartR` and a threshold of less than 10 base pairs of difference between sequences.”

Results:

“...and 0 loci that had similar sequences were excluded.”

*In Table 1 can the studbook numbers of the individuals included in the list of historical samples be provided?

The studbook number of individuals are provided in supplementary Table 3.

Supplementary Table 3. Studbook numbers of historical samples.					
No.	Studbook ID	Birth	End	Death/Transfer	Sample ID
1	135	SD-WAP USA	Yalooni OMAN	Transfer	OXL587
2	304	SD-WAP USA	Yalooni OMAN	Death	OXL561
3	319	Los Angeles USA	Yalooni OMAN	Death	OXL562
4	512	SD-WAP USA	Yalooni OMAN	Transfer	OXL564
5	652	Yalooni OMAN	Yalooni OMAN	Death	OXL566
6	656	Yalooni OMAN	Yalooni OMAN	b	OXL568
7	661	Yalooni OMAN	Yalooni OMAN	b	OXL570
8	826	Yalooni OMAN	Yalooni OMAN	Death	OXL573
9	1035	Yalooni OMAN	Yalooni OMAN	b	OXL575
10	1074	Yalooni OMAN	Yalooni OMAN	b	OXL577
11	1203	Yalooni OMAN	Yalooni OMAN	b	OXL578
12	1256	Yalooni OMAN	Yalooni OMAN	b	OXL581
13	1268	Yalooni OMAN	Yalooni OMAN	b	OXL582
14	1294	Yalooni OMAN	Yalooni OMAN	b	OXL583
15	1455	Yalooni OMAN	Yalooni OMAN	Death	OXL584
16	1470	Yalooni OMAN	Yalooni OMAN	Death	OXL585
17	40	Phoenix USA	Yalooni OMAN	Death	OXL693
18	1596	Al Ain UAE	NWRC Taif KSA	Death	OXL676
19	1600	Al Ain UAE	NWRC Taif KSA	Death	OXL675
20	1604	Al Ain UAE	NWRC Taif KSA	Transfer	OXL674
21	1154	Thumamah KSA	Thumamah KSA	b	OXL669
22	1155	Thumamah KSA	Thumamah KSA	b	OXL670

23	1158	Thumamah KSA	Thumamah KSA	b	OXL671
----	------	--------------	--------------	---	--------

*In Table 2, it would be helpful to include a column listing Ne and one for allelic richness.

Ne and allelic richness were added to Table 2.

Table 2. Summary statistics calculated in each group. Standard deviation is shown in parentheses.

Group	Number of individuals	Allelic richness	Shannon index	Observed heterozygosity	Expected heterozygosity	Inbreeding coefficient	Ne	Ne CI Low - CI high
Oman founder	10	1.268 (0.194)	0.386 (0.255)	0.263 (0.223)	0.268 (0.194)	0.023 (0.312)	16	15.2-16.8
USA founder	2	1.259 (0.281)	0.282 (0.301)	0.257 (0.325)	0.259 (0.278)	0.007 (0.371)	NA	NA
WWR Mix	89	1.281 (0.166)	0.431 (0.211)	0.274 (0.170)	0.281 (0.166)	0.022 (0.136)	22.4	22.2-22.7
WWR Oman	9	1.270 (0.195)	0.387 (0.255)	0.281 (0.235)	0.270 (0.195)	-0.027 (0.288)	44.3	39.4-50.4
WWR UAE	15	1.256 (0.193)	0.374 (0.261)	0.233 (0.199)	0.256 (0.193)	0.072 (0.292)	2.1	2-2.1

The lower the inbreeding coefficient values the lower the level of inbreeding.

Ci - confidence intervals. Ne in the USA founder group was not calculated due to the low number of individuals sampled) n=2.(

*L 285-6.(“ the main source of variation is mostly within individuals rather than between groups.”) I think the authors may mean to say “ the main source of variation is within groups, rather than between groups.”

The sentence was modified as suggested.

*One point that would provide perspective in the discussion and be useful for those considering further work is that the Rad-Seq data constitute an explicit data set and any further or expanded analysis would need to repeat work presented in this manuscript. This is in contrast to generating whole genome sequence data which can more readily be compared across experiments, investigators, and time. It would be very important to retain samples, especially perhaps, historical samples for genome sequencing analysis.

We agree with the reviewer opinion. We added the following text to acknowledge this important point:

“It is worth to mention that efforts to generate the reference genome of the Arabian Oryx are under way (personal communication Dr. Brooks; University of Florida). This important resource will make possible to map the SNPs used in this study, which will allow comparisons across experiments, investigators and time. We acknowledge that it is also important to retain samples, especially historical samples, for future analyses at the genome level.”

*The power of the Rad-Seq approach is impacted by sample size which reflects founder effects and bias of ascertainment.

To identify the impact of sample size on genetic statistics, standard deviation was calculated for all the statistics in Table 2.

Table 2. Summary statistics calculated in each group. Standard deviation is shown in parentheses.

Group	Number of individuals	Allelic richness	Shannon index	Observed heterozygosity	Expected heterozygosity	Inbreeding coefficient	Ne	Ne CI Low - CI high
-------	-----------------------	------------------	---------------	-------------------------	-------------------------	------------------------	----	---------------------

Oman founder	10	1.268 (0.194)	0.386 (0.255)	0.263 (0.223)	0.268 (0.194)	0.023 (0.312)	16	15.2-16.8
USA founder	2	1.259 (0.281)	0.282 (0.301)	0.257 (0.325)	0.259 (0.278)	0.007 (0.371)	NA	NA
WWR Mix	89	1.281 (0.166)	0.431 (0.211)	0.274 (0.170)	0.281 (0.166)	0.022 (0.136)	22.4	22.2-22.7
WWR Oman	9	1.270 (0.195)	0.387 (0.255)	0.281 (0.235)	0.270 (0.195)	-0.027 (0.288)	44.3	39.4-50.4
WWR UAE	15	1.256 (0.193)	0.374 (0.261)	0.233 (0.199)	0.256 (0.193)	0.072 (0.292)	2.1	2-2.1

The lower the inbreeding coefficient values the lower the level of inbreeding.

Ci - confidence intervals. Ne in the USA founder group was not calculated due to the low number of individuals sampled) n=2.(

*In Table 2, it is stated that two individuals from the “USA founders” of historical population were included in the analyses of genetic variation, yet in Table 1, it appears that no historical samples were used for generating the Rad-Seq. Could the authors please explain this apparent discrepancy.

Table 1 was modified to correct this mistake.

Table 1. Details of samples from the Historical ‘World Herd’ and the current WWR herd used for this study.

Group	Reserve/Zoo	Samples used for Mt-DNA analyses	Samples used for ddRAD analyses	Description
UK founder*	London	1	-	A male born in London RP (London Zoo) which originally came from ‘World Herd’ and translocated to KSA in 1989.
KSA founder*	Thumamah	5	-	These were translocated from the ‘World Herd’ in USA to KSA.
USA founder*	SD-WAP	4	1	San Diego Wildlife Safari Park, Escondido, USA.
	San Diego	1	-	San Diego Zoo, San Diego, USA.
	Phoenix	2	-	Phoenix Zoo, Phoenix, Arizona, USA.
	Brownsville	1	-	Gladys Porter Zoo, Brownsville, USA.
	Los Angeles	1	1	Arabian oryx ‘World Herd’ Los Angeles Zoo & Botanical Gardens.
Oman Founder*	Jalooni	18	10	Jalooni Reserve Office for Conservation Advisor, Muscat, Oman, Currently Office for Conservation of the Environment. These samples represented the founder of Omani Arabian oryx reintroduction project in 1980s.
UAE founder*	Al-Ain Zoo	3	-	Al-Ain Zoo, Al-Ain, UAE. These animals represent the founders’ of reintroduced oryx of Al-Ain Zoo in UAE.
WWR-Mix	WWR-Mix	108	88	WWR-Mix represent the offspring which results from crossbreeding of Oman-Herd and WWR-UAE which were translocated to the WWR in late 2011.
WWR-Oman	WWR-Oman	13	10	WWR-Oman Herd: represented oryx which retrieved from the wild during 1996 – 2007.
WWR-UAE	WWR-UAE	17	15	WWR-UAE Herd: represent the Arabian oryx introduced to WWR in 2011.
	Total**	174	125	

UK: United Kingdom; KSA: Kingdom of Saudi Arabia; UAE: United Arab Emirates; WWR: Al-Wusta Wildlife Reserve.

***Samples obtained from Marshall et al. (1999).**

****The difference in the number of samples between those use for mtDNA and RAD-seq was due the suitability and quality of DNA for downstream experiments or did not yield results suitable data.**

*Furthermore, since there will be impacts of diversity among the USA founders population depending on their founder contributions and pedigree relationships, can confidence estimates be established for the values in Table 2? The authors might use simulation methods to evaluate the variance in results that come from the Rad-Seq data and generate metrics of confidence and/or significance for the data presented in this study. Without some assessment of the confidence intervals or significance of their findings, the relevance to ongoing management is lessened.

Standard deviations were added to all the genetic statistics calculated.

Table 2. Summary statistics calculated in each group. Standard deviation is shown in parentheses.

Group	Number of individuals	Allelic richness	Shannon index	Observed heterozygosity	Expected heterozygosity	Inbreeding coefficient	Ne	Ne CI Low - CI high
Oman founder	10	1.268 (0.194)	0.386 (0.255)	0.263 (0.223)	0.268 (0.194)	0.023 (0.312)	16	15.2-16.8
USA founder	2	1.259 (0.281)	0.282 (0.301)	0.257 (0.325)	0.259 (0.278)	0.007 (0.371)	NA	NA
WWR Mix	89	1.281 (0.166)	0.431 (0.211)	0.274 (0.170)	0.281 (0.166)	0.022 (0.136)	22.4	22.2-22.7
WWR Oman	9	1.270 (0.195)	0.387 (0.255)	0.281 (0.235)	0.270 (0.195)	-0.027 (0.288)	44.3	39.4-50.4
WWR UAE	15	1.256 (0.193)	0.374 (0.261)	0.233 (0.199)	0.256 (0.193)	0.072 (0.292)	2.1	2-2.1

The lower the inbreeding coefficient values the lower the level of inbreeding.

Ci - confidence intervals. Ne in the USA founder group was not calculated due to the low number of individuals sampled) n=2.(

* In the mitochondrial DNA haplotype analysis, in the text (l 227-8), “Haplotype E had the lowest frequency (<1%) and was present in a single individual of the WWR-Mix group,” but as Figure 3 depicts was present in KSA, USA, and UAE populations.

With clarified this sentence and re-wrote it as follows:

“Haplotype E had the lowest frequency (<1%) and was present in a single individual of the WWR-Mix group, however, this haplotype was present in the was present in KSA, USA, and UAE groups.”

Here is an example of how the manuscript’s focus is on the animals in Oman. The perspective might be enlarged, as this is the first and most thorough study of its kind, to reflect the role of the population in the Sultanate of Oman in the global efforts to preserve the genetic variation of this endangered species.

I appreciate the work that went into this study and hope that the findings can be clarified and published.

Appendix C

Associate Editor Comments to Author (Professor Steve Brown):

Comments to the Author:

The manuscript is much approved and can be accepted after dealing with the minor suggestions of the reviewer

Reviewer comments to Author:

Reviewer: 3

Comments to the Author(s)

Review – Royal Society RSOS-210558.R1

Overall comments to authors:

- The aim of this paper is to use population genetic tools, using data obtained from one mitochondrial gene and genome-wide SNP genotyping completed using ddRAD-sequencing, to describe the levels of genetic diversity and genetic management history of the Arabian oryx in Oman. The authors use a number of analytical techniques to calculate multiple population genetic measures and provide implications for future genetic management based on the results of past management.
- The authors have done an excellent job responding to the suggestions and edits provided by the four reviewers in the first review phase. Parts of the analyses are more clear and it is evident that much work has gone into adding to the discussion to make the results relevant and applicable to future management. The main suggestion is that much of the new content in the Introduction, Discussion, Tables, and Supplementary information needs more editing for clarity, ease of reading, and for more detail. A number of suggestions have been provided below in the section-by-section comments.

We are grateful with the reviewer and commend his/her effort in pointing out the many details we missed.

Section by Section comments:

Abstract

- Line 21: It is unclear what “WWR” program is at this point in abstract.

After referring the first time to Al-Wusta Wildlife Reserve, we added the acronym WWR:

“The breeding program at the Al-Wusta Wildlife Reserve (WWR) in Oman...”

- Line 24: Need space before sentence starting “We...”

Space was added.

- Line 26: “three different groups” could have more explanation. Groups of what?

The word individuals was added to the sentence and now reads:

“Inference of ancestry and spatial patterns of SNP variation show the presence of three ancestral sources and three different groups of individuals”.

- Line 28: should read “contribute to maximizing genetic diversity.”

The word was changed as suggested:

“We identified individuals and groups that could most effectively contribute to maximizing genetic diversity Introduction”.

- Line 46-47: Awkward phrasing of sentence “captive breeding was not aimed to minimize...”. Rephrase to say something like “captive breeding was not implemented using individual genetic management through minimizing population mean kinship”.

The phrasing was changed as suggested:

“captive breeding was not implemented using individual genetic management through minimizing population mean kinship”.

- Lines 52-56: The reviewer appreciates the addition of supplementary Table 1 and the detail put into this. However, at this point in the introduction it is still unclear what “low levels of genetic diversity” and “high inbreeding and relatedness” mean to the authors. While a number of papers have been added and cited and the reader is suggested to review the Supplemental Table, it would be very helpful to the reader to be given some of the numbers to understand the context of what is meant by low and high in previous literature here in the introduction.

Examples of levels of genetic diversity and inbreeding were added to the text:

“More recent microsatellite studies of current captive breeding programs in Jordan, the United Arab Emirates (UAE) and Qatar found relatively low levels of genetic diversity (e.g. observed heterozygosity ranging from 0.332 to 0.535), some differentiation between herds or groups and a high level of inbreeding and relatedness (e.g. $F_{IS}=0.073-0.431$; Supplementary Table 1; 6, 7, 10, 11, 12, 13)”.

- Line 56: Rephrase start of this sentence to read something like “In addition to the research completed using microsatellites, 12 mtDNA control region haplotypes....”.

The sentence was rephrased as suggested:

“In addition to the research completed using microsatellites, 12 mtDNA control region (mtDNA CR) haplotypes”

- Line 57: include reference to Supplementary Table 1 in addition to Supplementary Table 2.

Reference to supplementary table 1 was added:

“...were identified in the Arabian oryx populations (Supplementary Tables 1 and 2; 11, 12, 15)”.

- Line 85: add reference/citation about random mating and maintenance of genetic diversity

The following citation was added to the text:

Caballero, A., Rodríguez-Ramilo, S., Avila, V., & Fernández, J. (2010). Management of genetic diversity of subdivided populations in conservation programmes. *Conservation Genetics*, 11(2), 409-419.

- Line 87-88: Difficult to understand phrasing “has acknowledge the importance...”, unsure of the aim of this sentence.

The sentence has been rephrased to improve clarity and now reads:

“This opportunity has been identified by the WWR management and consequently taken action to evaluate the current genetic variation and the genetic contribution of the different population sources to maximise the genetic diversity of the Arabian oryx”.

Materials and Methods

- Line 95-96: Please check reference/citation, as there are errors in the citation software and how these are showing to the reader.

The reference has been fixed.

Results

- Line 221: this should be broken up into two sentences. “This haplotype...” is the start of a new sentence.

The sentence was broken in two sentences:

“Haplotype E had the lowest frequency (<1%) and was present in a single individual of the WWR-Mix group. This haplotype was also present in the KSA, USA, and UAE groups”.

- Line 231 and 260: Please check reference/citation, as there are errors in the citation software and how these are showing to the reader.

The two references have been fixed.

- Line 292-293: It is unclear what “above analyses” were altered with the changed assignments after re-assigning the three incorrectly identified individuals. In addition, it might make more sense in the flow of results to put this result earlier in this section so that you can confirm that all downstream analyses (whatever they may be!) were done with these corrected identifications.

This sentence was changed to refer to “all downstream analyses” and the paragraph relocated to the beginning of the Results section as suggested. The sentence now reads:

“After confirming with WWR management, these samples were assigned to the correct groups for all the downstream analyses”.

Discussion

- Line 339: wording is awkward, should read “current sequencing technologies allow for genotyping thousands of markers...”

The sentence was changed as suggested and now reads:

“Furthermore, SNP data provides a more accurate representation of the genome, because current sequencing technologies allow for genotyping thousands of markers compared to tens of markers for typical microsatellites datasets”.

- Line 349-351: The authors present a summary of their SNP results and demonstration of an increase in heterozygosity in the WWR-Mix group and say that this is an interesting result. However, there is not interpretation as to why this interesting or not expected. Please provide more discussion on this point.

We added a sentence highlighting the importance of this result:

“This result is encouraging as it shows that a management based on random mating can conserve genetic diversity at adequate levels”.

- Line 361-362: wording “history of frequent founder effects” – this phrasing is a bit confusing. Please clarify what you mean here related to admixture and movement of individuals related to the q-profile and FST results.

We removed our mention of FST from this sentence, as we believe this measure of genetic differentiation is not related with this result. We added also the expected result from a history of frequent founder effects. The sentence now reads:

“This observation is probably a reflection of the species’ history of frequent founder effects which generally result in a depletion of genetic diversity”.

- Line 364-365: Awkward sentence structure or not a sentence.

Sentence was rephrased and now reads:

“Furthermore, we observe that most alleles, including private alleles from the founder groups, are present in the WWR-Mix”.

- Line 365-367: These sentences also need to be edited. End sentence after “our results demonstrate the advantage of using the q -profile”, then provide examples in the follow up sentences.

Sentences were edited as suggested and now read:

“Our results demonstrate the advantage of using the q -profile. For instance, we observe that genetic variation measured by heterozygosity is relatively uniform between groups, except in the USA founder group”.

- Line 381: missing word? “such as the data generated in this study?”

The word data was added to the sentence:

“such as the data generated in this study”.

- Line 382-384: should read “clear strategy aimed at maximizing genetic diversity and evolutionary potential and minimizing the effects of inbreeding”.

The sentence was edited as suggested:

“With the support of genetic data, such as the data generated in this study, the breeding program at the WWR could move from a random breeding approach to one with a clear strategy aimed at maximizing genetic diversity and evolutionary potential and minimizing the effects of inbreeding”.

Other

- Figure 6: This was difficult to read as there were black bars in the pdf version over the x and y axes

The Figure was reformatted. To ensure that figures will be displayed properly, we will submit the original files in PowerPoint format.

- Figure 7: Visually, the x-axis is off to the left a little bit making the ID numbers not quite line up in the heat map. Is the black bar in the figure meant to be there? Might be good to remove that if possible.

The Figure was reformatted. To ensure that figures will be displayed properly, we will submit the original files in PowerPoint format.

- Table 2: Would be nice to format this table so that headings are not split across multiple rows as well as so the values in each row. This table is very difficult to read as it is currently formatted. In addition, the Table description/header is missing.

Table was formatted as suggested and the description was added:

- Table 3: The last column says that all p-values are greater than 0.01. Did you mean less than? If so the sign needs to be reversed, and if you do mean greater than, you should provide the value. In addition, the Table description/header is missing.

The sign less than was corrected and the description was added:

Table 2. Summary statistics calculated in each group. Standard deviation is shown in parentheses.

Group	Sample size	Allelic richness	Shannon index	Ho	He	FIS	Ne	Ne CI Low - CI high
Oman founder	10	1.268(0.194)	0.386(0.255)	0.263(0.223)	0.268(0.194)	0.023(0.312)	16	15.2-16.8
USA founder	2	1.259(0.281)	0.282(0.301)	0.257(0.325)	0.259(0.278)	0.007(0.371)	NA	NA
WWR Mix	89	1.281(0.166)	0.431(0.211)	0.274(0.170)	0.281(0.166)	0.022(0.136)	22.4	22.2-22.7
WWR Oman	9	1.270(0.195)	0.387(0.255)	0.281(0.235)	0.270(0.195)	-0.027(0.288)	44.3	39.4-50.4
WWR UAE	15	1.256(0.193)	0.374(0.261)	0.233(0.199)	0.256(0.193)	0.072(0.292)	2.1	2-2.1

The lower the inbreeding coefficient values the lower the level of inbreeding.

Ho – observed heterozygosity; *He* expected heterozygosity; *FIS* – inbreeding coefficient; *Ne* – effective population size. Ci - confidence intervals. *Ne* in the USA founder group was not calculated due to the low number of individuals sampled (n=2).

- Supplementary Figure 1: the “k” in the table of haplotypes is lowercase and all other letters are uppercase.

The capital K was added to the figure:

Haplotype Label	3	9	1	1	1	1	2	2	2	2	2	3	3	3	3	3	4	4	4	5	5	6	
	5	4	1	3	9	7	1	3	6	1	5	3	0	7	6	4	6	9	0	3	9	5	7
Hap_A	G	C	T	C	G	A	A	A	T	G	C	C	C	T	T	G	C	A	T	T	C	A	C
Hap_B	G	C	T	T	G	A	A	G	C	G	C	C	C	T	T	G	C	A	T	T	C	A	T
Hap_C	G	C	T	T	G	A	A	G	C	G	C	C	C	T	T	G	C	A	T	T	T	A	T
Hap_D	G	C	T	T	G	A	A	A	T	G	T	T	C	T	C	A	C	A	T	T	C	A	C
Hap_E	G	C	T	T	G	A	G	A	T	G	T	T	C	T	C	A	C	A	T	T	T	A	C
Hap_F	G	C	C	T	G	A	A	A	T	G	C	C	C	T	T	A	C	A	C	T	C	G	C
Hap_G	G	C	C	T	G	G	A	A	T	G	C	C	C	T	T	A	C	A	C	T	C	G	C
Hap_H	G	C	C	T	G	A	A	A	T	G	C	C	C	T	T	A	C	G	C	T	C	G	C
Hap_I	G	T	T	T	A	G	G	A	T	A	C	T	T	C	T	A	T	A	T	C	C	A	C
Hap_J	G	C	T	T	G	A	G	A	T	G	C	T	C	T	C	A	C	A	T	T	T	A	C
Hap_K	A	T	T	T	A	G	G	A	T	A	C	T	T	C	T	A	T	A	T	C	C	A	C
Hap_L	A	T	T	T	A	G	G	A	T	A	C	T	T	C	T	A	T	A	T	T	C	A	C

- Supplementary Figure 2: the figure legend has the X- and Y-axis label information reversed in the description.

The legends for both axes were corrected:

Supplementary Figure 2. Bar chart of mtDNA CR haplotypes (A-G) among the current population of Al-Wusta Wildlife Reserve (WWR). The Y-axis displays the number of individuals in each haplotype, the X-axis display the haplotype (A-G). The colours represent the animals' herd (Blue: WWR-Mix, Brown: WWR-Oman herd and Green: WWR-UAE Herd).

- Supplementary Figure 6: would be nice to add the same figure legend and more detailed description here that is in the paper Figure 7, particularly since the individual IDs will be difficult or impossible to read in this figure, and might be blurry when zooming in, the reader will still want to get as much information out of the figure as possible.

The Figure description was improved as suggested:

Supplementary Figure 6. Heatmap of the probabilities of identity by descent in the in all the contemporary individuals (WWR-Mix, WWR-Oman and WWR-UAE). As described by Endelman & Jannink (68), each diagonal element has a mean that equals $1+f$, where f is the inbreeding coefficient (*i.e.* the probability that the two alleles at a randomly chosen locus are IBD from the base population). As this probability lies between 0 and 1, the diagonal elements range from 1 to 2. Because the inbreeding coefficients are expressed relative to the current population, the mean of the off-diagonal elements is $-(1+f)/n$, where n is the number of loci. Yellow and red colours indicate those individuals that are more related to each other. The identification number of each individual is shown in the margins of the figure, where the last letter denotes whether the individual is male (M) or female (F).

- Supplementary Figure 7: The caption for this figure could provide more details about how these individuals were originally determined to be assigned to the incorrect group. What was the process that was used to identify these individuals?

The caption of the figure was extended to provide a clearer explanation about how individuals were designated to their correct group.

Supplementary Figure 7. Estimates of relatedness and results from the program STRUCTURE were used to determine the correct group of individuals that were labelled incorrectly. In each case, a small explanation is provided to explain the reason individuals were mislabelled.

- Supplementary Figure 8: The caption for this figure could provide more details so that I can stand alone. What is meant here by three management herds (enclosures)? The reader needs to go find the details in the main paper to understand what the three herds are in relationship to the WWR-Mix herd so more detail is needed.

The caption of the figure was improved, so the reader can understand the figure without relying on the text.

“Supplementary Figure 8. Principal component analysis (PCA) using the first two principal components of individuals sampled at Al-Wusta Wildlife Reserve (WWR). Each point represents one individual, and the colour of the point represents the enclosure in which the individual was located at the time of sampling. The figure shows that clusters of individuals are not determined by the enclosure in which individuals are located. This observation indicates that genetic patterns presented in this study were not due to drift resulting from the isolation between the three enclosures”.

Reviewer: 2

Comments to the Author(s)

I have read the revised manuscript by Rawahi et al. and find that it has been substantially improved compared to the original submission. The authors have done an excellent job in providing thorough and satisfactory responses to the reviewers' comments, including my own, and have revised the manuscript accordingly. The streamlined text and corrected writing have improved the readability and presentation of the manuscript. I recommend this first-rate manuscript be accepted for publication.

We thank the reviewer for his/her valuable comments on the manuscript.

Reviewer: 4

Comments to the Author(s)

The revision responds adequately to the comments of reviewers.

We appreciate the reviewer's time and attention which greatly improved the manuscript.